# LEARNING IDENTITY-PRESERVING TRANSFORMATIONS ON DATA MANIFOLDS

## ABSTRACT

Many machine learning techniques incorporate identity-preserving transformations into their models to generalize their performance to previously unseen data. These transformations are typically selected from a set of functions that are known to maintain the identity of an input when applied (e.g., rotation, translation, flipping, and scaling). However, there are many natural variations that cannot be labeled for supervision or defined through examination of the data. As suggested by the manifold hypothesis, many of these natural variations live on or near a low-dimensional, nonlinear manifold. Several techniques represent manifold variations through a set of learned Lie group operators that define directions of motion on the manifold. However theses approaches are limited because they require transformation labels when training their models and they lack a method for determining which regions of the manifold are appropriate for applying each specific operator. We address these limitations by introducing a learning strategy that does not require transformation labels and developing a method that learns the local regions where each operator is likely to be used while preserving the identity of inputs. Experiments on MNIST and Fashion MNIST highlight our model's ability to learn identity-preserving transformations on multi-class datasets. Additionally, we train on CelebA to showcase our model's ability to learn semantically meaningful transformations on complex datasets in an unsupervised manner.

## 1 INTRODUCTION

A goal of many machine learning models is to accurately identify objects as they undergo natural transformations – a task that humans are adept at. According to the manifold hypothesis, natural variations in high-dimensional data lie on or near a low-dimensional, nonlinear manifold (Fefferman et al., 2016). Additionally, the manifolds representing different classes are separated by low density regions (Rifai et al., 2011a). Natural physical laws govern the possible transformations that objects can undergo and many of the identity-preserving transformations (e.g., changes in viewpoint, color, and lighting) are shared among classes of data. Sharing of transformations between classes enables increased efficiency in defining data variations – a model can represent a limited set of transformations that can describe a majority of variations in many classes. Several machine learning models incorporate specific identity-preserving transformations that are shared among a large number of classes to generalize the performance of their model to unseen data. These include equivariant models that incorporate transformations like translation and rotation into intermediate network layers (Cohen & Welling, 2016; Cohen et al., 2018) and data augmentation techniques that apply known identity-preserving variations to data while training (Cubuk et al., 2019; Ho et al., 2019; Lim et al., 2019; Sohn et al., 2020; He et al., 2020; Chen et al., 2020). However, many datasets have natural transformations shared among classes that are not easily prespecified from intuition, making it critical that we develop a model that can learn both 1) a representation for these transformations without explicit transformation labels and 2) the context for which locations in the data space each transformation is likely to be relevant.

Manifold learning strategies estimate the low-dimensional manifold structure of data. A subset of these techniques learn to transform points on the manifold through nonlinear Lie group operators (Rao & Ruderman, 1999; Miao & Rao, 2007; Culpepper & Olshausen, 2009; Sohl-Dickstein et al., 2010; Cohen & Welling, 2014; Hauberg et al., 2016; Connor & Rozell, 2020; Connor et al., 2021). Lie group operators represent infinitesimal transformations which can be applied to data

through an exponential mapping to transform points along a manifold, and a manifold can be globally defined by a set of operators that each move in different directions along it (Hoffman, 1966; Dodwell, 1983). A Lie group operator model is well-suited for representing natural data variations because the operators can be learned from the data, applied to data points to transform them beyond their local neighborhoods, and used to estimate geodesic paths.

While the Lie group operator models have many benefits, previous approaches demonstrate the two shortcomings noted above. First, to learn Lie group operators that represent a data manifold, pairs of training points are selected which lie within a neighborhood of one another. The training objective encourages efficient paths between these nearby points and the choice of training point pairs influences the types of manifold transformations that are learned. Recent papers incorporating Lie group operators into machine learning models have either used predefined operators that represent known transformations groups (e.g., the 3D rotational group $SO(3)$ (Falorsi et al., 2019)), required transformation labels for selecting point pairs when training (Connor & Rozell, 2020), or randomly selected pairs of points from the same class (Connor et al., 2021). To learn an effective model with datasets having no labeled transformation structure, we require a point pair selection strategy that identifies points that are related through the transformations the model aims to learn. Second, existing Lie group operator models have lacked a method for determining which regions of the manifold are appropriate for each operator, meaning that every operator is equally likely to be used at every point on the manifold. This is a flawed assumption because, while many transformations are shared between classes, there are also data variations that are unique to specific data classes. Additionally, in a dataset with several manifolds (each representing a different class), there is a limited extent to which a transformation can be applied without moving a point onto another manifold.

The main contributions of this paper are the development of methods to address the two critical shortcomings of Lie group operator-based manifold models noted above. Specifically, motivated by finding perceptually similar training samples without transformation labels, we first introduce a point pair selection strategy to learn a manifold representation of natural variations shared across multiple data classes without requiring transformations labels. Second, we develop a method that uses a pretrained classifier (measuring identity-preservation of transformed samples) to learn the local regions where each operator is likely to be used while preserving the identity of transformed samples. This approach enables us to analyze the local structure of the data manifold in the context of the learned operators and to describe the invariances of the classifier. We demonstrate the efficacy of these strategies in the context of the Manifold Autoencoder (MAE) model (Connor & Rozell, 2020) to learn semantically meaningful transformations on MNIST (LeCun et al., 1998), Fashion MNIST (Xiao et al., 2017), and CelebA (Liu et al., 2015).

## 2 BACKGROUND

**Manifold Learning**   Manifold learning models estimate the low-dimensional structure of high-dimensional data by utilizing the property that local neighborhoods on the manifold are approximately linear. Traditional techniques represent the manifold through a low-dimensional embedding of the data points (Tenenbaum et al., 2000; Roweis & Saul, 2000; Belkin & Niyogi, 2003; Maaten & Hinton, 2008) or through estimates of linear tangent planes that represent local directions of manifold motion (Dollár et al., 2007; Bengio & Monperrus, 2005; Park et al., 2015). While these traditional manifold learning approaches are useful for understanding low-dimensional data structure, in many cases the input data space is an inefficient representation of the data. For example, data in the pixel space suffers from the curse of dimensionality and cannot be smoothly interpolated while maintaining identity (Bengio et al., 2005). Many approaches have used neural networks to learn a low-dimensional latent space in which manifold models can be incorporated. The contractive autoencoder (CAE) estimates manifold tangents by minimizing the the Jacobian of the encoder network, encouraging invariance of latent vectors to image space perturbations (Rifai et al., 2011c;b;a; Kumar et al., 2017). Several methods estimate geodesic paths in the latent space of a trained variational autoencoder (VAE) model (Arvanitidis et al., 2018; Chen et al., 2018; Shao et al., 2018; Arvanitidis et al., 2019) and use this approach to learn VAEs with priors that are estimated using the Riemannian metrics computed in the latent space (Arvanitidis et al., 2021; Kalatzis et al., 2020).

**Lie Group Operators**   A Lie group is a group of continuous transformations which also defines a manifold by representing infinitesimal transformations that can be applied to input data (Hoffman,

1966; Dodwell, 1983). Several methods incorporate Lie groups into neural networks to represent data transformations that are identity-preserving within the model (Cohen & Welling, 2014; Cohen et al., 2018; Cosentino et al., 2021). A prevalent strategy is to learn a dictionary of Lie group operators that are mapped to a specific group element through the matrix exponential $\text{expm}(\cdot)$ (Rao & Ruderman, 1999; Ham & Lee, 2006; Miao & Rao, 2007; Culpepper & Olshausen, 2009; Sohl-Dickstein et al., 2010; Cohen & Welling, 2014; Connor & Rozell, 2020; Connor et al., 2021). In these models, each operator $\mathbf{\Psi}_m$, called a *transport operator*, describes a single direction along the manifold and is parameterized by a single coefficient $c_m$. Given an initial data point $\mathbf{z}$, the transport operators define a generative model where transformations can be derived from sampling sparse coefficients $c_m \sim \text{Laplace}(0, \zeta)$:

$$\widehat{\mathbf{z}} = \text{expm}\left(\sum_{m=1}^{M} \mathbf{\Psi}_m c_m\right) \mathbf{z} + \epsilon, \tag{1}$$

where $\epsilon \sim \mathcal{N}(0, \sigma_\epsilon^2 \mathbf{I})$.

The manifold autoencoder (MAE) incorporates the transport operator model into the latent space of an autoencoder to learn a dictionary of operators that represent the global, nonlinear manifold structure in the latent space (Connor & Rozell, 2020). This model has been shown to effectively learn reusable operators with transformation supervision, and it will provide the context for demonstrating the effectiveness of the methods developed in this paper.

**Disentanglement** One class of methods that also aim to identify factors of variation in data is disentanglement methods which learn features that each vary one independent characteristic of the data (Higgins et al., 2018). In a supervised learning scenario, there are many disentangling techniques that separate data style from content by encouraging similarity between the content features in samples from the same class (Tenenbaum & Freeman, 2000; Reed et al., 2014; Mathieu et al., 2016; Bouchacourt et al., 2018) or encouraging features to map to known class (Cheung et al., 2015) or transformation labels (Hinton et al., 2011; Kulkarni et al., 2015; Yang et al., 2015). There are also techniques that can disentangle latent representations without labels like InfoGAN (Chen et al., 2016) and $\beta$-VAE (Higgins et al., 2017). The goal of our work is distinct from disentanglement work because, rather than identifying independently varying factors of variation, we aim to learn non-linear operators that correspond to transformations on the data manifold (which may not vary independently). Our approach is advantageous because it can faithfully represent longer transformation paths, learn variations while maintaining image reconstruction, and change the number of learned variations by increasing or decreasing the number of learned operators.

## 3 METHODS

The MAE learns a low-dimensional latent representation of the data by defining an encoder function $f : \mathcal{X} \to \mathcal{Z}$ that maps high-dimensional data points $\mathbf{x} \in \mathbb{R}^D$ to low-dimensional latent vectors $\mathbf{z} \in \mathbb{R}^d$ and a decoder function $g : \mathcal{Z} \to \mathcal{X}$ that maps the latent vectors back into the data space (Connor & Rozell, 2020). Transport operators $\mathbf{\Psi}$ are incorporated into the latent space to learn manifold-based transformations. Before learning the transport operators, the autoencoder is pretrained to extract a latent representation of the data using the traditional autoencoder reconstruction objective.

After pretraining, the autoencoder weights are fixed and the operators are trained with the following objective, which encourages the learning of transport operators that generate efficient paths between the latent vectors $\mathbf{z}_0$ and $\mathbf{z}_1$ (coinciding with $f(\mathbf{x}_0)$ and $f(\mathbf{x}_1)$) that are nearby on the manifold:

$$\mathcal{L}_\Psi = \frac{1}{2}\left\|\mathbf{z}_1 - \text{expm}\left(\sum_{m=1}^{M} \mathbf{\Psi}_m c_m\right)\mathbf{z}_0\right\|_2^2 + \frac{\gamma}{2}\sum_m \|\mathbf{\Psi}_m\|_F^2 + \zeta\|\mathbf{c}\|_1, \tag{2}$$

where $\gamma, \zeta > 0$.

Objective (2) is minimized via an alternating minimization scheme. Specifically, at each training iteration, points pairs $\mathbf{x}_0$ and $\mathbf{x}_1$ are selected in the input space and encoded into the latent space $\mathbf{z}_0$ and $\mathbf{z}_1$. Then, coefficients $\mathbf{c}$ are inferred between the encoded latent vectors (by minimizing (2) with respect to $\mathbf{c}$) to estimate the best path between $\mathbf{z}_0$ and $\mathbf{z}_1$. After inference, the coefficients are

fixed and a gradient step is taken to minimize (2) with respect to transport operator weights. Once learned, these operators represent different types of motion that traverse the manifold and they can be combined to generate natural paths on the manifold.

After fitting transport operators to the latent space of a fixed autoencoder network, there is a final fine-tuning training phase that updates the network weights and transport operators simultaneously using a joint objective that combines the autoencoder reconstruction loss with $\mathcal{L}_\Psi$. This fine-tuning step addresses the potential of a mismatch between the data manifold and the learned latent structure by adapting the latent structure to fit the transport operators learned between the selected training point pairs. We empirically find that breaking up training into these three phases increases the stability with which the transport operators can be learned. Given the context of this MAE model, in the following sections we describe our contributions.

### 3.1 Unsupervised Transformation Learning

There are many possible strategies for selecting training point pairs for a Lie group operator model and the choice of a strategy can dictate the types of learned transformations. Point pairs may be selected as random samples from the dataset or from within the same class. These points, however, are likely to be outside of local manifold neighborhoods and may not provide representations of the natural, perceptually smooth variations. Point pairs can also be selected as nearest neighbors in the pixel or autoencoder latent space (denoted with $h(\mathbf{x}) = \mathbf{x}$ and $h(\mathbf{x}) = f(\mathbf{x})$ in (3), respectively):

$$\mathbf{x}_1 = \underset{\mathbf{x} \in \mathcal{D}}{\arg\min} \|h(\mathbf{x}_0) - h(\mathbf{x})\|_2^2. \tag{3}$$

However, semantic transformations of interest may not be exhibited through pixel similarity nor through unsupervised autoencoder feature similarity. In some applications there is additional information that can aid in point pair selection, such as rotation angle, semantic label, or time index in a temporal sequence. However, for most complex datasets, information about the transformations of interest is not available. This necessitates a strategy for selecting training point pairs which can be used to learn natural transformations without requiring additional transformation supervision.

We generalize transport operator training to incorporate an unsupervised learning strategy that can be applied to a wide array of datasets. In particular, we assume access to a classifier pretrained on a large-scale dataset (likely taken from a model zoo) with similar statistics as our training dataset of interest (Nguyen et al., 2020). In this approach, we select point pairs by setting $h(\mathbf{x})$ from (3) equal to the penultimate layer of the classifier, similar to the previously proposed perceptual loss (Johnson et al., 2016). It has been shown that high-level features in such classifier networks correspond to semantic characteristics of the data and finding points nearby in this feature space can result in image pairs that are perceptually similar without being exactly the same (Johnson et al., 2016; Yosinski et al., 2015). Incorporating the classifier features as a supervision strategy for point pair selection enables our model to learn transport operators without requiring transformation labels or class labels. We show that this metric leads to learned operators that correspond to semantically identifiable transformations. Discussion of different point pair strategies can be found in Appendix E.

### 3.2 Learning Local Transport Operator Statistics to Encourage Identity Preservation

A trained transport operator model provides a dictionary of operators that can be applied globally across the entire data space. This model has flexibility to define local manifold characteristics through the coefficients $\mathbf{c}$ which specify the combination of operators to apply to a given point. The standard prior on the coefficients is a factorial Laplace distribution parameterized by a single scale parameter $\zeta$, which encourages transformations to be defined by a sparse set of operators:

$$p_\zeta(\mathbf{c}) = \prod_{m=1}^{M} \frac{1}{2\zeta} \exp\left(-\frac{|c_m|}{\zeta}\right). \tag{4}$$

By setting a fixed prior on coefficients across the entire manifold, there is an implicit assumption that the manifold structure is the same over the whole dataset and every operator is equally likely to be applied at every point. However, this is a flawed assumption because not all transformations

in one class of data are present in all other classes and within classes there may be regions with different manifold structure. Failing to capture the local statistics of transport operator usage could result in transformed points that depart from the data manifold and change their identity.

To address this limitation, we introduce a coefficient encoder network which maps latent vectors to scale parameters that specify the Laplace distribution on the coefficients at those latent points. The goal of this network is to estimate local coefficient distributions that maximize the identity-preservation of points (as measured by a pretrained classifier) transformed with operators characterized by sampled coefficients as in (1). Given labeled observations $(\mathbf{z}, y)$ and a pretrained classifier on the latent space $r(\cdot)$, we aim to learn a network $q_\phi(\mathbf{c}|\mathbf{z})$ that outputs parameters that can be used to produce augmented samples $\widehat{\mathbf{z}}$ without changing the classifier output $r(\widehat{\mathbf{z}}) = y$.

To maximize the likelihood of augmented input data maintaining the same classifier output, we adapt the concept of consistency regularization from semi-supervised learning (Bachman et al., 2014; Sohn et al., 2020). In our context, we have a pretrained classifier and a dictionary of transport operators and we aim to find a distribution on the coefficients at individual input points that result in consistent classification outputs when $T_\Psi(\mathbf{c})$ is applied. The specific objective can be chosen from a variety of loss functions that encourage similarity in classifier probability outputs. We specifically minimize the KL-divergence between the classifier output $r(\mathbf{z})$ and the estimated probability of the transformed output $r(\widehat{\mathbf{z}})$.

Unfortunately, the consistency regularization objective can be trivially minimized by setting $\mathbf{c} = 0$, resulting in an identity transformation and the same classifier output. Therefore, we want to encourage the largest coefficient values possible while maintaining the identity of our initial data point. This motivates the addition of a KL-divergence regularizer that encourages the distribution $q_\phi(\mathbf{c}|\mathbf{z})$ to be similar to a specified Laplace prior with a fixed scale $\zeta$ like in (4). Our final objective for training the coefficient encoder network is: (a more detailed derivation can be found in Appendix A)

$$E = D_{KL}(r(\mathbf{z})\|r(\widehat{\mathbf{z}})) + D_{KL}\left(q_\phi(\mathbf{c}|\mathbf{z})\|p_\zeta(\mathbf{c})\right). \tag{5}$$

The coefficient encoder introduces a principled way to build identity-preservation directly into our model and to identify local manifold characteristics throughout the latent space. Additionally, it can quantify the extent with which operators can be applied in different parts of the latent space without changing class prediction, indicating transformations to which the classifier is invariant. Likewise, points or region that have small encoded coefficient scale weights indicate closeness to class boundaries and can undergo only small transformations without changing identity.

### 3.3 Decreasing Computational Complexity of Coefficient Inference

To compute the gradient for the matrix exponential during training and inference, previous works have used a linear approximation (Rao & Ruderman, 1999), learned the operators in a factored form (Sohl-Dickstein et al., 2010), or used analytic gradients that are not favorable for parallel computations (Culpepper & Olshausen, 2009; Connor & Rozell, 2020; Connor et al., 2021). The main computational bottleneck for learning transport operators is the coefficient inference step, where (2) is minimized with respect to $\mathbf{c}$. The $\ell_1$ norm term is non-smooth, leading to slower convergence when using subgradients found via automatic differentiation (Boyd & Vandenberghe, 2004). In this work we employ a forward-backward splitting scheme (Beck & Teboulle, 2009) that alternates between automatic differentiation of the matrix exponential term, and proximal gradients for the $\ell_1$ norm (Parikh & Boyd, 2014). This significantly speeds up the training process and allows for scaling up the learning of transport operators to complex datasets. To the best knowledge of the authors, this is the first work to demonstrate learning of Lie group operators on datasets as large and complex as CelebA. Additional details on the inference algorithm are provided in Appendix B.

## 4 Analysis

For our experiments, we examine the ability of our proposed approaches to enable the MAE model to learn natural transformations of datasets where the underlying identity-preserving transformations are not easily identifiable. We also highlight the benefits of incorporating the coefficient encoder network that captures the transport operator local usage statistics by encoding coefficient scale values that best maintain the identity of latent vectors.

We work with three datasets: MNIST (LeCun et al., 1998), Fashion MNIST (Xiao et al., 2017), and CelebA (Liu et al., 2015). We select MNIST and Fashion MNIST because they contain several classes that share natural transformations but they do not have transformation labels. We select CelebA to highlight our ability learn natural transformations in a larger, more complex dataset. As a classic dataset used in papers that aim to disentangle dataset features (Higgins et al., 2017; Chen et al., 2016; Hu et al., 2018; Lin et al., 2019), CelebA contains semantically meaningful natural transformations that may be amenable to qualitative labeling.

In all experiments, we followed the general training procedure put forth previously (Connor & Rozell, 2020) by separating the network training into three phases: the autoencoder training phase, the transport operator training phase, and the fine-tuning phase. The fine-tuning phase adapts the autoencoder latent space to fit the learned transformations in the event that that pretrained latent space does not match the desired data manifold. We select training point pairs that are nearest neighbors in the feature space of the final, pre-logit layer of a ResNet-18 (He et al., 2015) classifier pretrained on ImageNet (Russakovsky et al., 2015). After completely training the MAE, we fix the autoencoder network weights and transport operator weights and train the coefficient encoder network with the objective derived in Section 3.2. Additional details on the datasets, network architectures, and training procedure are available in the Appendix.

We compare against the contractive autoencoder (CAE) (Rifai et al., 2011c) and $\beta$-VAE (Higgins et al., 2017), two other methods for incorporating data structure into the latent space.. The CAE represents another technique that learns a manifold representation in a neural network latent space. In their model, the manifold is represented by estimated tangent planes at latent point locations. The $\beta$-VAE learns to disentangle factors of variation along latent dimensions through an increase in the weighted penalty on the KL-divergence term in the VAE objective. We choose these methods because they also learn natural dataset variations in the latent space without transformation labels.

### 4.1 LEARNING NATURAL DATA VARIATIONS

First, we show how well the perceptual point pair selection strategy described in Section 3.1 enables the MAE model to learn natural data variations in datasets and we highlight the usefulness of a nonlinear manifold model for generating latent space paths. Fig. 1 shows the paths generated by transport operators trained in this model. The image in the middle column in each block of images is the reconstructed version of the input image $\mathbf{x}_0$. The images to the left and right of the middle show the reconstructed images generated by an individual learned operator applied to the encoded latent vector $\mathbf{z}_0$: $\mathbf{z}_c = \text{expm}(\mathbf{\Psi}_m c)\mathbf{z}_0$, $c = -N_c, ..., N_c$. We see an individual operator generates a similar transformation across multiple inputs, and in many cases, the transformations induced by the transport operators are semantically meaningful. We also show that the perceptual point pair selection strategy is effective over a range of datasets.

While many of the operators can be assigned a semantic label through qualitative visual inspection, the CelebA dataset has attribute labels for the images which enable a quantitative analysis of the connection between learned transport operators and dataset attributes. To classify attributes, we fine-tune a ResNet-18 pretrained on ImageNet with 16 classification heads for attributes including smile, beard, hair color, and pale skin (Mao et al., 2020). With this classifier, we are able to identify transport operators that correspond to specific dataset attributes. Fig. 2 shows the classification outputs of the attribute classifier for example transport operators. Our model learns operators that vary hair color and skin paleness as well as several others which are shown in the Appendix.

In Fig. 1 we compare transport operator paths to those generated by the CAE and the $\beta$-VAE. The CAE-generated paths are the directions of motion on the tangent planes estimated at individual points. The $\beta$-VAE paths are generated by varying the value of one latent dimension while the others remain fixed. While our method and the $\beta$-VAE learn several qualitatively similar transformations, our method is capable of doing so without significantly sacrificing reconstruction performance. In the Appendix, we include examples of the transformations generated by each learned operator.

A primary motivation for learning the data manifold is to generate new views of data that cannot arise from pre-defined functions. To highlight the ability of our model to generate new views that preserve identity, we compare both interpolated and extrapolated paths from the MAE to those from the CAE, $\beta$-VAE, and a traditional auto-encoder in Fig. 3. Interpolated paths offer insight in the smooth variations between two data points, while extrapolated paths show the capacity to generate

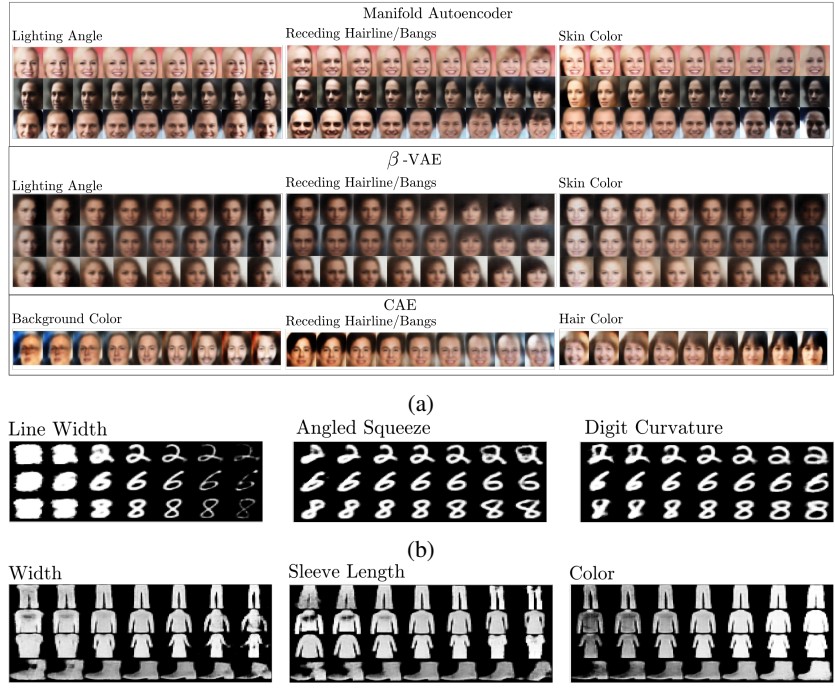

(a)

(b)

(c)

Figure 1: Paths generated by applying a subset of learned transport operators on three datasets. In each figure, images in the middle column of the image block are the reconstructed inputs and images to the right and left are images decoded from transformed latent vectors in positive and negative directions, respectively. (a) Comparing the transformations generated by three learned transport operators to transformations generated by $\beta$-VAE and CAE. The transport operators learn semantically meaningful transformations similar to the disentangled $\beta$-VAE representation while maintaining a higher resolution in image outputs. (b-c) Transport operators learned using the perceptual point pair selection strategy generate natural transformations on both MNIST and Fashion MNIST.

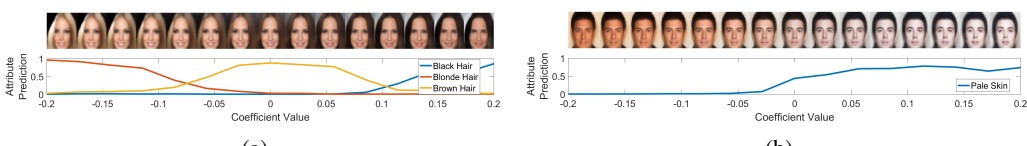

(a)                                                    (b)

Figure 2: Paths generated by the application of two learned transport operators with the associated attribute classifier probability outputs for the transformed images. Our model learns operators that correspond to meaningful CelebA dataset attributes like (a) hair color (b) pale skin.

new views of data. For MAE, the paths are estimated by inferring coefficients $\mathbf{c}^*$ between two latent points $\mathbf{z}_0$ and $\mathbf{z}_1$ and then generating the path: $\mathbf{z}_t = \mathrm{expm}\left(\sum_{m=1}^{M} \Psi_m c_m^* t\right) \mathbf{z}_0$. When the path multiplier $t$ is between 0 and 1 that indicates interpolation and path multipliers beyond 1 indicate extrapolation. For all other methods, Euclidean distance between features is used for both interpolation and extrapolation. In these figures, the first block of images corresponds to the interpolated path with the selected final point $\mathbf{x}_1$ surrounded in an orange box. The second block of images corresponds to the extrapolated paths.

To quantify the identity preservation of each transformation, we input each generated image into a pretrained classifier and plot the prediction of the correct class. All four methods perform interpolation effectively but our trained model estimates the extrapolated paths more accurately. Fig. 4 shows how that accuracy varies during extrapolation sequences for 4000 samples. The MAE is better at generating extrapolated outputs that maintain class identity. The lower classification accuracy in fashion MNIST is due to both a more challenging dataset and the lower resolution of autoencoder image outputs when compared to input images.

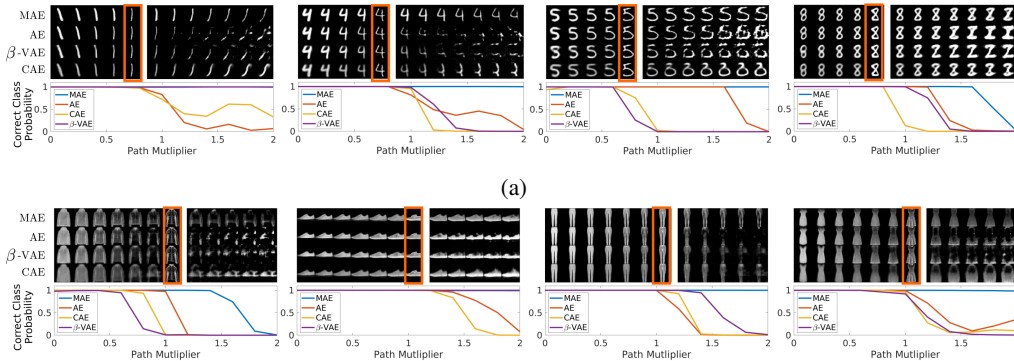

(a)

(b)

Figure 3: Identity preservation of transformed paths as quantified by a pretrained classifier output. In the figures on the top, the first block of images corresponds to the interpolated path with selected final point $\mathbf{x}_1$ surrounded in an orange box. The second block of images corresponds to the extrapolated paths. Below the images are plots of the probability of the class label associated with the inputs $\mathbf{z}_0$ and $\mathbf{z}_1$. A path multiplier between 0 and 1 indicates interpolation and path multipliers beyond 1 indicate extrapolation. (a) MNIST (b) Fashion MNIST.

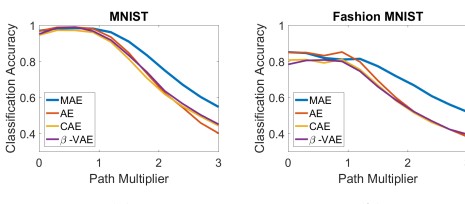

(a)                                         (b)

Figure 4: The average accuracy of the classifier output for images at each point along the interpolation/extrapolation sequence. When the path multiplier is between 0 and 1 that indicates interpolation and path multipliers beyond 1 indicate extrapolation. (a) MNIST (b) Fashion MNIST.

## 4.2 LEARNING LOCAL MANIFOLD STRUCTURE

After the MAE is trained, we have a dictionary of transport operators that describe manifold transformations and a network with a latent space that is adapted to the manifold. We then train the coefficient encoder network to estimate the coefficient scale weights as a function of points in the latent space. To visualize how the use of the transport operators varies over the latent space, we generate an Isomap embedding (Tenenbaum et al., 2000) of latent vectors and color each point by the encoded scale parameter for coefficients associated with each of the transport operators. Fig. 5a shows these embeddings for MNIST data. Each operator has regions of the latent space where their use is concentrated.

By training the coefficient encoder to maximize the similarity between the classification outputs for an input sample and for a transformed version of that sample, the network aids in identifying which transport operators can be applied to inputs in regions of the data space without changing the identity of the input. This helps significantly with data augmentation where the goal is to create new samples with in-class variations. To highlight this benefit, in Fig. 6 we show samples augmented by applying transport operators with randomly sampled coefficients to an input latent vector. In each block of images, the leftmost image (in a green box) is the input image and the images to the right are decoded augmentations. The top row shows samples augmented with transport operators controlled by coefficients sampled from Laplace distributions with encoded coefficient scale weights. The bottom row shows samples augmented with transport operators controlled by coefficients sampled from Laplace distributions with a fixed scale parameter. While both strategies generate some realistic variations of the data, using the encoded scale weights improves identity-preservation of the transformed output. The augmentations with the encoded scale weights are better at maintaining the identity of the sampled points.

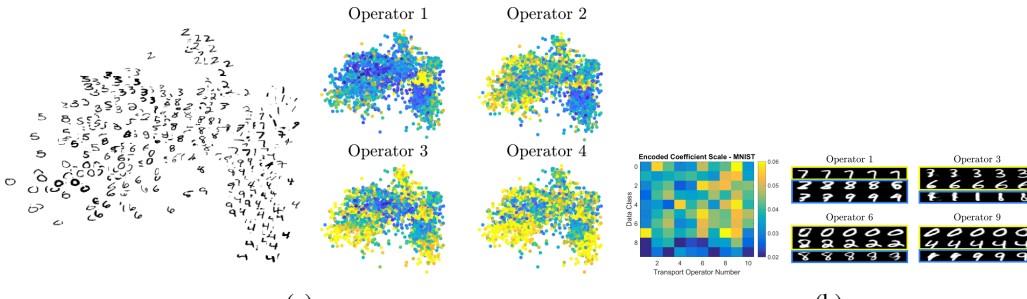

(a)                                                                    (b)

Figure 5: (a) Visualizations of the encoded coefficient scale weights. The leftmost image shows an Isomap embedding of the latent vectors with input images overlaid. The scatter plots on the right show the same Isomap embedding colored by the encoded coefficient scale weights for several operators (yellow indicates large scale weights and blue small scale weights). We see operators whose use is localized in regions of the manifold space. (b) The average coefficient scale weights for each class on each transport operator for MNIST. High scale weights for a given operator (yellow) indicate it can be applied to a given class without easily changing identity. The images on the right show examples of the operators applied to classes with high encoded scale weights (in the top yellow boxes) and classes with low encoded scale weights (in bottom blue boxes). The examples with low coefficient scale weights change classes more easily than other examples.

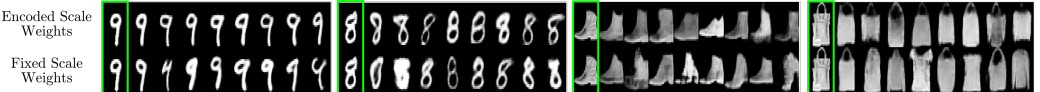

Figure 6: Examples of samples generated by transport operators using coefficients sampled with encoded scale weights (top row) and with a fixed scale weight (bottom row). Images in the green box are the input images and the remaining images in each row are transformed outputs.

Because the coefficient encoder uses a pretrained classifier to determine the coefficient distributions in the latent space, the resulting encoded scale weights can inform about the types of manifold transformations the classifier is invariant to. Fig. 5b shows the average coefficient scale weights for each class (rows) and each transport operator (columns) for MNIST. From this we can identify classes for which the classifier is both sensitive and robust to transformations, represented by small and large scale weights respectively. We can also examine which classes share the use of the same transformations. The images to the right in Fig. 5b show transport operators being applied to samples with high encoded scale weights (in a yellow box) and samples with low encoded scale weights (in a blue box). We can examine the characteristics of transformations that are better suited to some classes than others. For instance, operator 3 in Fig. 5b increases the curve at the bottom of digits. This is a natural transformation for classes 3, 5, and 6 which all have higher coefficient scale weights for this operator, but when this is applied to a 1, that makes it look like an 8.

## 5 CONCLUSION

In this work we develop methods to improve the effectiveness and utility of Lie group operator models for learning manifold representations of datasets with complex transformations that cannot be labeled. We do this by introducing a perceptual point pair selection strategy for training operators and by developing a method that uses a pretrained classifier to learn local regions where operators are likely to be used while preserving the identity of transformed samples. We demonstrate the efficacy of our approach in learning natural dataset variations with the MAE. While this is a powerful model, users should be mindful of the biases that are introduced through training with specific supervision methods when drawing conclusions about learned data transformations. This work presents a promising technique for learning representations of natural data variations that can improve model robustness. In future work we can address some limitations by expanding this work to consider methods for localizing manifold structure with limited or no class labels and improving the reconstruction fidelity by applying the model within more complex generative model embeddings.

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

## Supplementary Material

## A  Coefficient Encoder Details

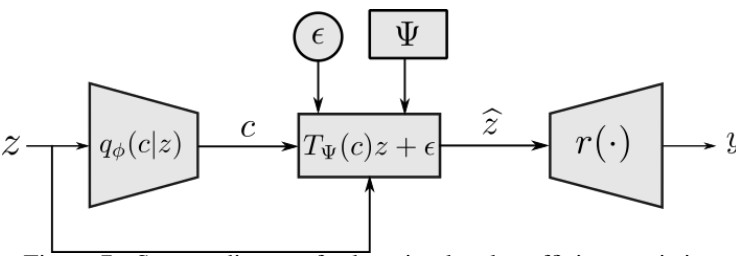

Figure 7:  System diagram for learning local coefficient statistics.

Our motivation in training an encoder to learn the coefficient statistics is to understand the regions of the manifold in which specific transport operators are applicable and the invariances of a pretrained classifier to the learned transport operator-induced transformations. In this section, we will outline a derivation of our coefficient encoder training objective from a variational inference perspective by learning a variational posterior for the coefficients using a deep neural network (Kingma & Welling, 2013; Rezende et al., 2014). Given observations $(\mathbf{z}_i, y_i)$ for $i = 1, \ldots, N$, and a pre-trained classifier $r(\cdot)$, we aim to learn a distribution $q_\phi(\mathbf{c}|\mathbf{z})$ from which we can sample to define the transformation $T_\Psi(\mathbf{c})$ that can be applied to $\mathbf{z}$ while maintaining class identity. In other words, we want the augmented $\widehat{\mathbf{z}} = T_\Psi(\mathbf{c})\mathbf{z} + \epsilon$ to result in $r(\widehat{\mathbf{z}}) = y$.

To encourage identity-preservation of vectors transformed with transport operators that are controlled by sampled coefficients, we maximize the likelihood of a class output for an observation under the system model diagrammed in Figure 7. This system connects the observed latent vector $\mathbf{z}$ to an output $y$ through a transformation defined by $\mathbf{c}$. We follow the derivation used in Rezende & Mohamed (Rezende & Mohamed, 2016) which introduces the variational posterior to estimate the log likelihood of the data.

Consider a parameterized distribution for the conditional likelihood of our observations:

$$\log p_\theta(y|\mathbf{z}) = \log \mathbb{E}_{p_\zeta(\mathbf{c})}\left[p_\theta(y|\mathbf{c}, \mathbf{z})\right] \tag{6}$$

$$= \log \int_{\mathbf{c}} p_\zeta(\mathbf{c})p_\theta(y|\mathbf{c}, \mathbf{z})d\mathbf{c} \tag{7}$$

$$= \log \int_{\mathbf{c}} q_\phi(\mathbf{c}|\mathbf{z})\frac{p_\zeta(\mathbf{c})}{q_\phi(\mathbf{c}|\mathbf{z})}p_\theta(y|\mathbf{c}, \mathbf{z})d\mathbf{c} \tag{8}$$

$$\geq \int_{\mathbf{c}} q_\phi(\mathbf{c}|\mathbf{z})\log\left[\frac{p_\zeta(\mathbf{c})}{q_\phi(\mathbf{c}|\mathbf{z})}p_\theta(y|\mathbf{c}, \mathbf{z})\right]d\mathbf{c} \tag{9}$$

$$= E_{q_\phi}\left[\log p_\theta(y|\mathbf{c}, \mathbf{z})\right] - D_{KL}\left(q_\phi(\mathbf{c}|\mathbf{z})|p_\zeta(\mathbf{c})\right) \tag{10}$$

$$= E_{q_\phi}\left[\log E_\epsilon p_\theta(y|\mathbf{c}, \mathbf{z}, \epsilon, \widehat{\mathbf{z}})\right] - D_{KL}\left(q_\phi(\mathbf{c}|\mathbf{z})|p_\zeta(\mathbf{c})\right). \tag{11}$$

In equation 6 and equation 7, we marginalize over the coefficients $\mathbf{c}$ as required under our system model in Figure 7. In equation 9, we lower bound the conditional likelihood with a variational lower bound derived from Jensen's inequality. Finally, we use the reparameterization trick to define $\widehat{\mathbf{z}}$ as a deterministic function of $\mathbf{c}$, $\mathbf{z}$, and the parameter-free random variable $\epsilon$ in equation 11.

To sample from $q_\phi(\mathbf{c} \mid \mathbf{z})$ when computing the expectations, we first use the reparameterization trick (Connor et al., 2021; Kingma & Welling, 2013; Rezende et al., 2014) to define the sampled coefficients $\widehat{\mathbf{c}}$ as a function of a uniform random variable $\mathbf{u} \sim \text{Unif}\left(-\frac{1}{2}, \frac{1}{2}\right)^M$:

$$\widehat{\mathbf{c}} = l_\phi(\mathbf{u}, \mathbf{z}) = -h_\phi(\mathbf{z})\,\text{sgn}(\mathbf{u})\log(1 - 2\,|\mathbf{u}|), \tag{12}$$

where $l_\phi$ is a defined mapping from a uniform distribution to a Laplace distribution and the Laplace scale parameters are defined by the output of the coefficient encoder $h_\phi$ that we aim to learn.

Using the reparameterization of $\widehat{\mathbf{c}}$ and $\widehat{\mathbf{z}}$, we can define the expectation:

$$E_{q_\phi}\left[\log E_\epsilon p_\theta(y|\mathbf{c}, \mathbf{z}, \epsilon, \widehat{\mathbf{z}})\right] = E_{\mathbf{u}}\left[\log E_\epsilon p_\theta(y|\widehat{\mathbf{c}} = l_\phi(\mathbf{u}, \mathbf{z}), \epsilon, \widehat{\mathbf{z}})\right] \tag{13}$$

$$\approx \frac{1}{JK}\sum_{j=1}^{J}\log\sum_{k=1}^{K} p_\theta\left(y|\widehat{\mathbf{c}}^{(j)} = l_\phi(\mathbf{u}^{(j)}, \mathbf{z}), \epsilon^{(k)}, \widehat{\mathbf{z}}\right). \tag{14}$$

Our pre-trained classifier defines $p_\theta$ which allows us to specify final objective using of $r(\cdot)$. Using a KL-divergence as the likelihood function between the ground truth labels $y_i$ and the classifier output for the augmented inputs $r(\widehat{\mathbf{z}}_i^{(j)})$ and simplifying the model by setting $\epsilon$ to 0, we get:

$$\log p_\theta(y|\mathbf{z}) \geq -\frac{1}{J}\sum_{j=1}^{J} D_{KL}\left(y_i|r(\widehat{\mathbf{z}}_i^{(j)})\right) - D_{KL}\left(q_\phi(\mathbf{c}|\mathbf{z})|p_\zeta(\mathbf{c})\right), \tag{15}$$

where in practice we use a single sample $J = 1$ and we estimate $D_{KL}\left(y_i|r(\widehat{\mathbf{z}}_i^{(j)})\right) \approx D_{KL}\left(r(\mathbf{z}_i^{(j)})|r(\widehat{\mathbf{z}}_i^{(j)})\right)$ for test samples for which we may not have class labels. These practical simplifications result in the objective in 5 which we want to minimize. The KL divergence between the $q_\phi(\mathbf{c}|\mathbf{z})$ and $p_\zeta(\mathbf{c})$ has a closed form expression (Gil, 2011):

$$D_{KL}\left(q_\phi(\mathbf{c}\mid\mathbf{z})\|p_\zeta(\mathbf{c})\right) = \log(h_\phi(\mathbf{z})) - \log(\zeta) + \frac{\zeta}{h_\phi(\mathbf{z})} - 1 \tag{16}$$

## B  COEFFICIENT INFERENCE DETAILS

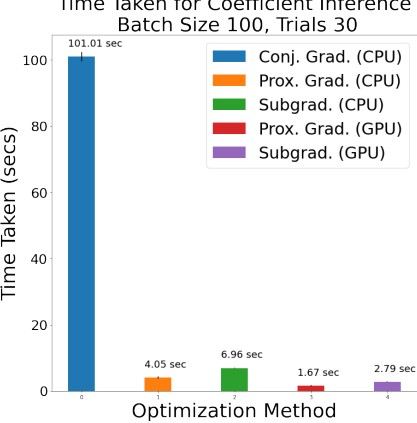

Figure 8: Comparison in time taken for various coefficient inference algorithms on fully trained transport operators. Each inference trial is performed over random batches of 100 point pairs drawn from the CelebA dataset (with a latent dimension of 32). 30 trials are run with the same $\mathbf{c}_0$ and tolerance. We used the proximal gradient (GPU) method.

To perform coefficient inference, previous Lie group operator methods (Connor & Rozell, 2020; Connor et al., 2021) use the analytic gradient proposed in (Culpepper & Olshausen, 2009) with a conjugate gradient descent solver. This method requires an eigenvalue decomposition for computing the gradient which is not favorable for parallel computations. Alternatively, we use the PyTorch implementation of the matrix exponential that allows for automatic differentiation. Furthermore, to handle the non-smooth $\ell_1$ norm in objective (2), we apply a proximal gradient step in a forward-backward splitting scheme (Beck & Teboulle, 2009). For the $\ell_1$ norm, the proximal gradient has a known, closed-form solution in the soft-thresholding function (Parikh & Boyd, 2014):

$$\mathcal{T}_\lambda(\mathbf{c}) = \text{sign}(\mathbf{c}) * \max\left(|\mathbf{c}| - \lambda, 0\right) \tag{17}$$

Let $\nabla_{\mathbf{c}} \frac{1}{2} \left\| \mathbf{z}_1 - \text{expm} \left( \sum_{m=1}^{M} \boldsymbol{\Psi}_m c_m \right) \mathbf{z}_0 \right\|_2^2 \approx \nabla \widetilde{f}(\mathbf{c})$ be a numerical approximation to the gradient of the $\ell_2$ term, found through automatic differentiation. Given initial coefficient values $\mathbf{c}_0$ drawn from an isotropic Gaussian with variance $4 \times 10^{-4}$, our gradient descent step is:

$$\mathbf{c}_{k+1} = \mathcal{T}_{\zeta \alpha_k} \left( \mathbf{c}_k - \alpha_k \nabla \widetilde{f}(\mathbf{c}_k) \right) \tag{18}$$

These steps are iterated upon until either a max iteration count is reached, or the change in coefficients, $\|\mathbf{c}_{k+1} - \mathbf{c}_k\|_2$, falls below some threshold. For all experiments we use a max iteration count of 800 and a tolerance of $1 \times 10^{-5}$. For our step-size, we use $\alpha_k = (0.985)^k \alpha_0$ with $\alpha_0 = 1 \times 10^{-2}$. We experimented with different acceleration methods (Kingma & Ba, 2015; Beck & Teboulle, 2009), and found that in many cases they resulted in worse performance.

The main parameter to select for inference is the sparsity-inducing parameter on the coefficient prior, $\zeta$. This value is sensitive to the dataset, latent dimension, and $\gamma$ hyper-parameter value. In practice, we fix a $\gamma$ value and pick $\zeta$ to be as high as possible, inducing sparsity in the inferred coefficients, while maintaining good reconstruction performance of $\widehat{\mathbf{z}}_1$. If $\zeta$ is too large, that will result in all coefficients going to zero, preventing proper reconstructive performance. On the other hand, setting $\zeta$ to be too small will result in all operators being used to represent the transformation between each point pair, preventing any meaningful structure from being learned during training. Parameters used in specific training runs are included in their respective Appendix sections.

When compared against the coefficient inference implementation from (Connor & Rozell, 2020), we perform inference for a batch of 100 samples in an average of 1.67 seconds over 100 trials whereas the inference from (Connor & Rozell, 2020) took an average of 101.01 seconds. We also compare the speed-up over a baseline that strictly uses automatic differentiation (subgradients) and compare the benefits from moving to GPU hardware in Figure 8. We use the proximal gradient (GPU) method from this figure. Experiments were run on a machine with an Intel i7-6700 CPU with 4.00 GHz and a Nvidia TITAN RTX.

## C  TRAINING STRATEGY

In all experiments, we follow the general training procedure put forth previously (Connor & Rozell, 2020). We train the MAE with three training phases: the autoencoder training phase, the transport operator training phase, and the fine-tuning phase. During the autoencoder training phase, the network weights are updated using a reconstruction loss objective: $E_{\text{AE}} = \|\mathbf{x} - \hat{\mathbf{x}}\|_2^2$.

During the transport operator training phase, the network weights are fixed and the transport operators are trained between pairs of points using the objective (2). Pairs of images $x_0, x_1$ are chosen using the perceptual point pair selection strategy described in Section 3.1. The images are then encoded into the latent space $z_0, z_1$. For each batch, the first step is to infer the coefficients between all pairs of latent vectors. Coefficient inference is best performed when the entries of the latent vectors are close to the range $[-1, 1]$. Because of this, we define a scale factor that can be applied to encoded latent vectors to reduce the magnitude of their entries prior to performing coefficient inference. In practice, we inspect the latent vector magnitudes after the autoencoder training phase and choose a scale that will adjust the magnitudes of the latent vector entries to be in the range $[-1, 1]$. This does not have to be a precise range for the latent vector magnitudes but instead is a practical guideline. Coefficient inference is performed on the scaled latent vectors as described in Sections 3.3 and Appendix B. After the coefficients are inferred for a batch, the weights on the dictionary elements are updated. This phase of training is performed until the loss values reach a plateau and the dictionary magnitudes plateau.

The fine-tuning training phase begins after the transport operator training phase is complete. In this phase, both the network weights and the transport operator weights are updated using the joint objective:

$$E = \lambda \left( \|\mathbf{x}_0 - \hat{\mathbf{x}}_0\|_2^2 + \|\mathbf{x}_1 - \hat{\mathbf{x}}_1\|_2^2 \right) + (1 - \lambda) E_{\Psi}, \tag{19}$$

where $E_{\Psi}$ is defined in (2). Using this objective, we alternate between taking steps on the transport operator weights while the network weights are fixed and taking steps on the network weights while the transport operator weights are fixed. Additionally, during the fine-tuning phase we incorporate

occasional steps in which we update the network weights using only the reconstruction loss to ensure effective image reconstruction.

In most cases, it is necessary to reduce the $\gamma$ parameter in front of the Frobenius norm dictionary regularizer prior to fine-tuning or the dictionary magnitudes will reduce to zero. We report the $\gamma$ we use for transport operator training and fine-tuning in the experimental details sections below. It may also be necessary to decrease the network learning rate during fine-tuning.

The coefficient encoder training requires a network that is trained to classify data from our selected dataset. We train this classifier using training data from a given dataset. For datasets with worse autoencoder reconstruction quality, we train the classifier in the latent space. Otherwise we train the classifier on images in the data space. With the MAE and classifier trained, we train the coefficient encoder network following the strategy described in Section 3.2 and Appendix A.

To train the CAE, we use the same autoencoder architecture used with the MAE with the addition of a Frobenius norm regularizer on the encoder Jacobian, weighted by a selected $\lambda$ value (different from the $\lambda$ in equation 19). The Jacobian is computed using PyTorch automatic differentiation. We find the Jacobian norm decreases to the same value irrespective of our choice of $\lambda$, leading us to choose $\lambda = 1$. For the $\beta$-VAE we use the same architectures outlined in (Higgins et al., 2017) with $\beta = 10$ for CelebA and $\beta = 5$ for MNIST and FMNIST. We find that setting $\beta$ any higher results in poor reconstructive performance. Scripts for training both comparison methods are included in the code repository.

Hyper-parameter tuning for all experiments was performed on the Georgia Tech Partnership for Advanced Computing Environment (PACE) clusters (PACE, 2017). Experiments were performed using a Nvidia Quadro RTX 6000. Runs training the CAE, and $\beta$-VAE on CelebA were all run on a separate machine with a Nvidia TITAN RTX.

## D   PARAMETER SELECTION

The MAE model has several hyperparameters that must be tuned and we will provide guidance to determining ideal parameter values for our experiments and future experiments. First we will describe some signs to look out for to identify if a run is succeeding or failing. One indicator that we compute is the transport operator difference which is:

$$E_{\Psi\text{diff}} = \frac{1}{2} \left\| \mathbf{z}_1 - \text{expm}\left( \sum_{m=1}^{M} \boldsymbol{\Psi}_m c_m \right) \mathbf{z}_0 \right\|_2^2 - \frac{1}{2} \left\| \mathbf{z}_1 - \text{expm}\left( \sum_{m=1}^{M} \widehat{\boldsymbol{\Psi}}_m c_m \right) \mathbf{z}_0 \right\|_2^2, \quad (20)$$

where $\widehat{\boldsymbol{\Psi}}_m$ is the dictionary after the gradient step is taken. A gradient step should decrease the transport operator objective meaning the $E_{\Psi\text{diff}}$ value should be positive for an effective step. If there are many gradient steps that result in negative $E_{\Psi\text{diff}}$ values, that indicates that the parameters are not optimal or that the learning rate is too large. This is an important metric to observe during fine-tuning to determine whether to select a smaller $\gamma$ or smaller network learning rate.

Signs of failure of a training run with selected training parameters:

- All operator magnitudes reduce towards zero.
- All inferred coefficients between point pairs are zero.
- Most of the operators generate transformation paths with latent values that increase quickly to infinity.
- Many steps have negative values for $E_{\Psi\text{diff}}$.
- The operator magnitudes increase which results in unstable training steps with NaN values in the computed objective.

**Dictionary regularizer parameter**   The dictionary regularizer parameter $\gamma$ is the weight on the Frobenius norm term in equation 2. This objective term serves two purposes. First, it balances the effect of the coefficient sparsity regularizer with the parameter $\zeta$. If $\zeta$ is large, the sparsity regularizer

encourages small coefficient magnitudes and one way to achieve that while still effectively inferring paths is to increase the magnitude of the operators. The Frobenius norm term must have a large enough influence to counterbalance this force or the operators will increase in magnitude to the point of being unstable. If a run is becoming unstable, we recommend decreasing $\zeta$ or increasing $\gamma$.

The second purpose of the dictionary regularizer term is to identify which operators are necessary to represent the transformations on the data manifold. If an operator is not being used to represent a transformation between $\mathbf{z}_0$ and $\mathbf{z}_1$ then the dictionary regularizer reduces its magnitude to zero. Therefore, during training we are able to estimate the model order based on how many dictionary elements remain non-zero. In our tests we vary $\gamma$ between $2 \times 10^{-8}$ and $2 \times 10^{-4}$. If $\gamma$ is too small, it will not counterbalance the coefficient sparsity term and the dictionaries will grow to unstable magnitudes. If $\gamma$ is too large it will reduce the magnitude of all operators to zero. During the fine-tuning steps, we have often found it necessary to reduce the $\gamma$ because, with a larger $\gamma$, both the operator magnitudes and the latent vector magnitudes can decrease substantially which leads to an ineffective manifold model.

**Coefficient sparsity parameter**   The coefficient sparsity parameter $\zeta$ in equation 2 controls the sparsity of the coefficients that are used to estimate paths between $\mathbf{z}_0$ and $\mathbf{z}_1$. We run tests with values of $\zeta$ between 0.005 and 2. From dataset to dataset the ideal value varies. If $\zeta$ values are too small then all of the operators are used to represent all of the paths between point pairs. The $\zeta$ should be increased so fewer than $M$ coefficients are used for each inferred path. When $\zeta$ is too large, all the coefficients go to zero during inference. This means there is no path inferred between $\mathbf{z}_0$ and $\mathbf{z}_1$ because of overweighting the sparsity constraint.

**Number of dictionary elements**   As mentioned above, the dictionary regularizer acts as a model order selection tool so our strategy for selecting number of dictionary elements $M$ is to increase the number of dictionary elements until some of their magnitudes begin reducing to zero during training. This indicates that some of the operators are not necessary for representing transformations.

**Latent dimension**   The latent dimension of the autoencoder is selected to ensure quality reconstructed image outputs.

**Relative weight of reconstruction and transport operator objectives**   The parameter $\lambda$ determines the weight of the reconstruction term relative to the transport operator term. We observe good performance of the model for $\lambda$ between 0.5 and 0.75.

## E   COMPARING POINT PAIR SELECTION STRATEGIES

In the past, transport operators have been trained using point pairs that were selected randomly from the same class (Connor et al., 2021) or with some knowledge of transformation labels (Connor & Rozell, 2020). In this work, we establish a method that learns natural transformations without requiring transformation labels using a perceptual point pair selection strategy. The perceptual loss metric (which compares the the features values from the penultimate layer of a pretrained classifier model) enables us to select point pairs that may share semantically meaningful transformations without being exactly the same. Fig. 9 shows examples of nearest neighbors selected using pixel similarity, similarity in the autoencoder latent space, and the perceptual loss metric. This highlights how the perceptual loss metric can be useful for identifying inputs with similar qualitative characteristics. For instance, in the set of images on the left, the perceptual loss metric identifies another bald man as a nearest neighbor which has similar characteristics of the initial image even though the exact image looks quite different.

When examining the success of learned operators, one characteristic we care about is how well the operators maintain the stability of generated paths. We consider generated paths stable if the latent vectors do not expand quickly to infinity. We have defined this as a useful characteristic for identity-preservation of applied operators – if operators expand latent vectors to infinity, that will very likely lead to a change in identity. Each of the operators can be viewed as the dynamics matrix of a continuous time linear dynamical systems model and we can analyze their stability as dynamical systems by observing their eigenvalues (Strogatz, 2000).

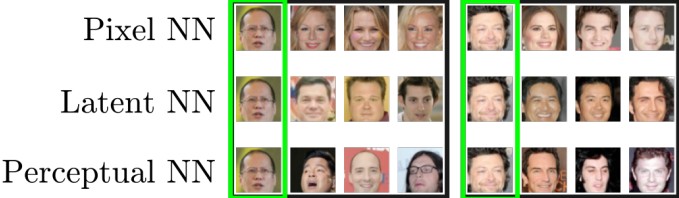

Pixel NN
Latent NN
Perceptual NN

Figure 9: Examples of nearest neighbors identified through pixel similarity, latent similarity, and the perceptual loss metric. In each set of 12 images, the images in the first column (contained in green boxes) are the initial reference images and the images to the right of those show three selected nearest neighbors. The perceptual loss metric identifies neighbors that share similar characteristics, like hair style, without being exactly the same.

In continuous time linear dynamical systems, there are three classifications of system stability. If a system is stable, the real parts of all eigenvalues are below zero. In a stable system, the magnitude of an input shrinks as time progresses forward. If a system is unstable, the real part of at least one eigenvalue is above zero. In unstable systems, the magnitude of an input expands to infinity as time progresses forward. Finally, if a system is marginally stable, then the magnitude of an input remains constant and the real parts of all of its eigenvalues are zero[1] (Strogatz, 2000). The transport operator model is unique because operators can represent dynamics matrices that define the natural data variations which do not have the same directionality as temporal systems (where time moves forward). Therefore transport operators can be applied with positive or negative coefficients and the traditional view of stability of temporal systems no longer applies.

To maintain the stability of generated paths, the operators need to generate cyclic transformation paths that neither increase nor decrease the latent vector magnitudes. This corresponds to marginal stability in linear dynamical systems. A marginally stable system has only imaginary eigenvalues with no real parts (Strogatz, 2000). Therefore, we can identify transport operators approaching marginal stability by investigating the magnitudes of the real parts of their eigenvalues. The maximum real magnitude of an eigenvalue of a dynamics matrix drives the speed with which the generated paths expand to infinity. Therefore this maximum real magnitude is the focus of our investigation into the stability of paths generated by learned operators. When the maximum magnitude of the real part for all eigenvectors associated with an operator is close to zero, that indicates that the operator is closer to marginal stability. Therefore, we quantify the stability of transport operator generated paths by looking at the maximum magnitude of real parts of eigenvalues associated with each transport operator. Fig. 10a shows the sorted maximum magnitude of the real parts of eigenvalues in each of the 16 operators learned in the experiment in Section 4.1 which learns natural MNIST variations. The blue x's are associated with the operators trained using the perceptual point pair selection strategy and the red dots are associated with the operators trained using a simple strategy of selecting point pairs as nearest neighbors in the latent space. With the exception of one operator, all the operators trained using latent space similarity for supervision have larger maximum magnitudes of real eigenvalue components than the operators trained using the perceptual loss metric. This indicates that the operators trained using the latent space similarity to select training point pairs are farther from marginal stability and can be seen as less stable by our definition of transport operator stability.

To view the effect of these learned operators more intuitively, we plot the values of the latent dimensions as individual operators are applied in Fig. 11. Each line in these plots shows the influence of the operators on a single latent dimension (Note that this experiment utilizes a latent dimension of 10). Fig. 11a shows the paths generated by transport operators trained with the perceptual point pair selection strategy . The plots with straight lines (i.e., operators 3, 4, 7, 12, 14, and 16) indicate that those operators had their magnitudes reduced to zero during training and therefore they have no impact on the latent vectors when applied. Most of the operators learned using the perceptual point pair selection strategy are close to cyclic except for operator 2 (the operator with the large real component magnitude in Fig. 10a). By contrast, several of the operators trained with point pairs selected

---

[1]Note that a system can also be marginally stable with at least one pair of eigenvalues with a zero real part and non-zero imaginary parts and the rest of the eigenvalues with non-positive real parts. For simplification of this analysis, we will focus on striving for systems which have eigenvalues with all real parts equal to 0.

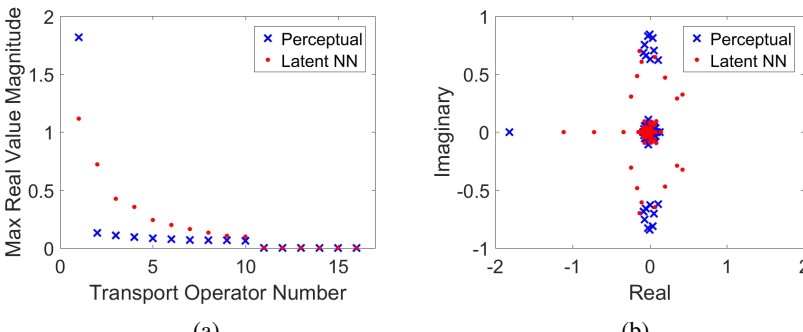

(a)                                                (b)

Figure 10: Analysis of eigenvalues of operators learned in the MNIST experiment. (a) The maximum absolute value of the real parts of eigenvalues computed from each learned operator. (b) Plot of the real and imaginary parts of the eigenvalues for each operator.

as nearest neighbors in the latent space extend to infinity (Fig. 11b). This can explain the larger maximum real value magnitudes in Fig. 10a. This analysis leads us to conclude that the perceptual point pair selection strategy has a greater potential to yield identity-preserving transport operators.

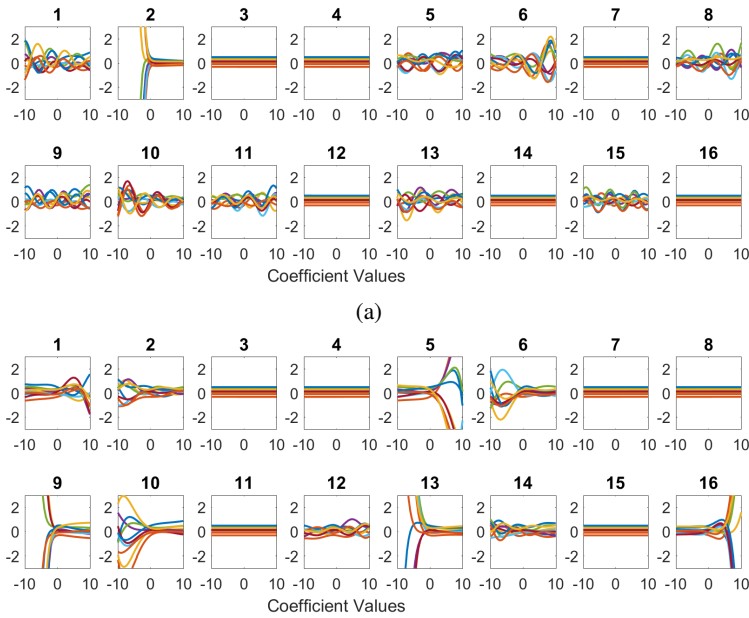

Figure 11: Visualizations of the effect of each learned transport operator on each dimension of an encoded latent vector. In each plot, each line represents a single latent dimension and the coefficient magnitude of the transformation varies on the x-axis. (a) Paths from operators learned with perceptual point pair supervision. (b) Paths from operators learned with point pairs selected as nearest neighbors in the latent space.

## F   MNIST EXPERIMENT DETAILS

The MNIST dataset is made available under the terms of the Creative Commons Attribution-Share Alike 3.0 license. We split the MNIST dataset into training, validation, and testing sets. The training set contains 50,000 images from the traditional MNIST training set. The validation set is made up of the remaining 10,000 images. The traditional MNIST testing set is used for our testing set. The input images are normalized so their pixel values are between 0 and 1. The network architecture used for the autoencoder is shown in Table 1. The training parameters for the transport operator training phase and the fine-tuning phase are shown in Tables 2 and 3.

Prior to training the coefficient encoder for the MNIST dataset, we train a classifier on the labeled MNIST image data which we use to encourage identity-preservation during coefficient encoder training. The training parameters for the coefficient encoder are shown in Table 4. The image classifier we use is based on the simple LeNet architecture with two convolutional layers and three fully connected layers (LeCun et al., 1998).

Table 1: Network Architecture for MNIST and Fashion MNIST Experiments

| Encoder Network | Decoder Network |
| --- | --- |
| Input $\in \mathbb{R}^{28 \times 28}$ | Input $\in \mathbb{R}^2$ |
| conv: chan: 64 , kern: 4, stride: 2, pad: 1 | Linear: 3136 Units |
| BatchNorm: feat: 64 | ReLU |
| ReLU | convTranpose: chan: 64, kern: 4, stride: 1, pad: 1 |
| conv: chan: 64, kern: 4, stride: 2, pad: 1 | BatchNorm: feat: 64 |
| BatchNorm: feat: 64 | ReLU |
| ReLU | convTranpose: chann: 64, kern: 4, stride: 2, pad: 2 |
| conv: chan: 64, kern: 4, stride: 1, pad: 0 | BatchNorm: feat: 64 |
| BatchNorm: feat: 64 | ReLU |
| ReLU | convTranpose: chan: 1, kernel: 4, stride: 2, pad: 1 |
| Linear: 2 Units | Sigmoid |

Table 2: Training parameters for the transport operator training phase of the MNIST experiment

| MNIST Transport Operator Training Parameters |
| --- |
| batch size: 250 |
| autoencoder training epochs: 300 |
| transport operator training epochs: 50 |
| latent space dimension ($z_{dim}$): 10 |
| $M : 16$ |
| $lr_{\text{net}} : 10^{-4}$ |
| $lr_{\Psi} : 10^{-3}$ |
| $\zeta : 0.1$ |
| $\gamma : 2 \times 10^{-6}$ |
| initialization variance for $\Psi$: 0.05 |
| number of restarts for coefficient inference: 1 |
| nearest neighbor count: 5 |
| latent scale: 30 |

Table 3: Training parameters for the fine-tuning phase of the MNIST experiment

| MNIST Fine-tuning Parameters |
| --- |
| batch size: 250 |
| transport operator training epochs: 100 |
| $lr_{\text{net}} : 10^{-4}$ |
| $lr_{\Psi} : 10^{-3}$ |
| $\zeta : 0.1$ |
| $\gamma : 2 \times 10^{-6}$ |
| $\lambda$: 0.75 |
| number of network update steps: 50 |
| number of $\Psi$ update steps: 50 |

Table 4: Training parameters for the MNIST Coefficient Encoder

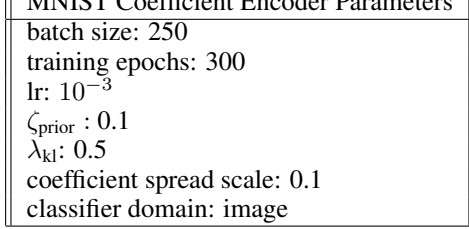

| MNIST Coefficient Encoder Parameters |
| --- |
| batch size: 250 |
| training epochs: 300 |
| lr: $10^{-3}$ |
| $\zeta_{\text{prior}} : 0.1$ |
| $\lambda_{\text{kl}}$: 0.5 |
| coefficient spread scale: 0.1 |
| classifier domain: image |

## G   MNIST EXPERIMENT ADDITIONAL RESULTS

Here we show additional experimental details and results for the MNIST experiment. Fig. 12 shows the magnitude of all 16 operators after the fine-tuning phase. Six of the operators have their magnitudes reduced to zero. Fig. 13 shows the paths generated by transport operators trained on MNIST data.

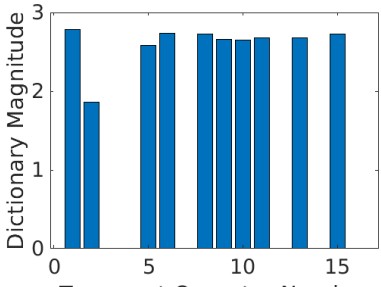

Figure 12: The magnitudes of the learned operators after fine-tuning.

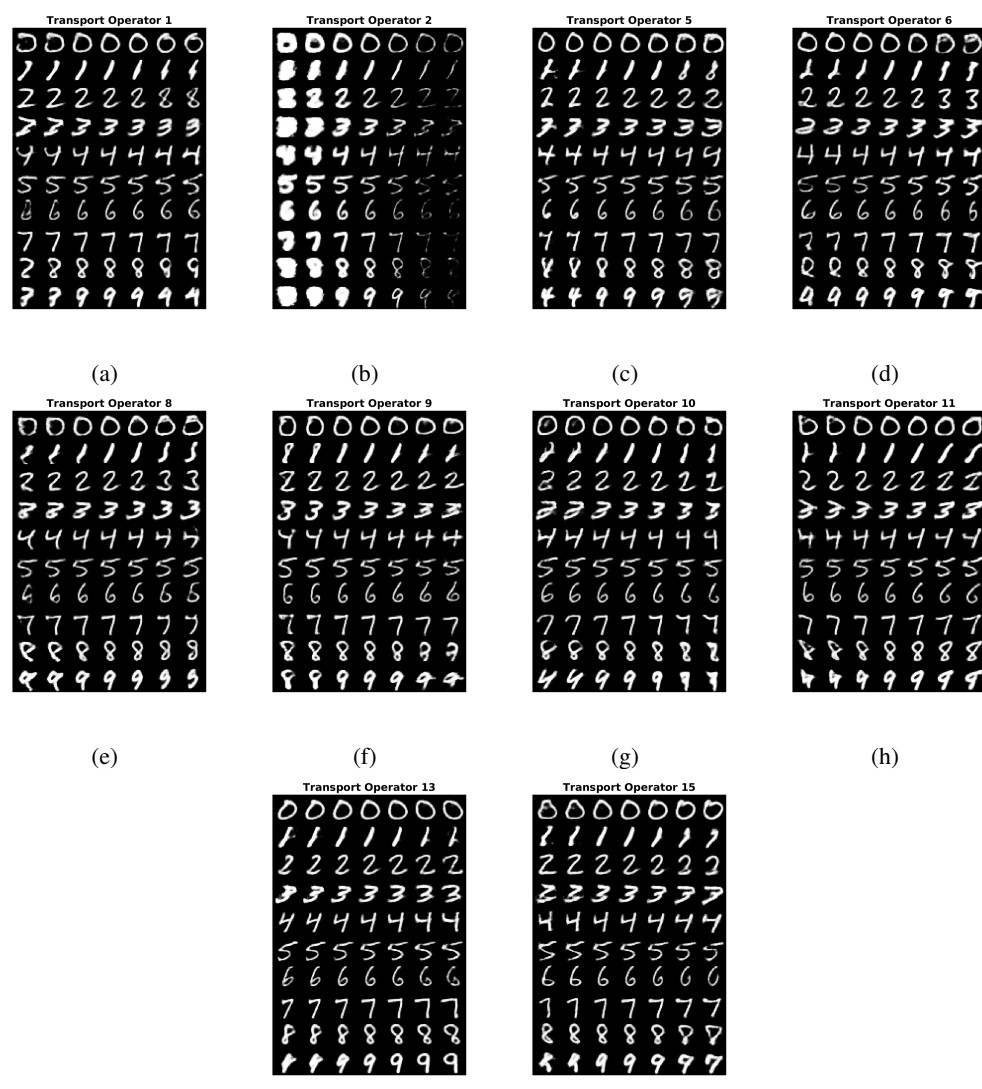

Figure 13: Paths generated by all non-zero transport operators trained on the MNIST dataset. Images in the middle column of the image block are the reconstructed inputs and images to the right and left are images decoded from transformed latent vectors in positive and negative directions, respectively

## H  FASHION-MNIST EXPERIMENT DETAILS

The Fashion-MNIST dataset is made available under the terms of the MIT license. We split the Fashion MNIST dataset into training, validation, and testing sets. The training set contains 50,000 images from the Fashion-MNIST training set. The validation set is made up of the remaining 10,000 images. The traditional Fashion-MNIST testing set is used for our testing set. The input images are normalized so their pixel values are between 0 and 1. The network architecture used for the autoencoder is the same as in the MNIST experiment and it is shown in Table 1. The training parameters for the transport operator training phase and the fine-tuning phase are shown in Tables 5 and 6.

Prior to training the coefficient encoder for the Fashion-MNIST dataset, we train a classifier the latent vectors associated with labeled Fashion-MNIST data which we use to encourage identity-preservation during coefficient encoder training. The training parameters for the coefficient encoder are shown in Table 7. The latent vector classifier has a simple architecture of Linear(512), ReLU, Linear(10), Softmax.

Table 5: Training parameters for the transport operator training phase of the Fashion-MNIST experiment

| Fashion-MNIST Transport Operator Training Parameters |
|---|
| batch size: 200 |
| autoencoder training epochs: 300 |
| transport operator training epochs: 50 |
| latent space dimension ($z_{dim}$): 10 |
| $M$ : 16 |
| $lr_{\text{net}} : 10^{-4}$ |
| $lr_{\Psi} : 10^{-3}$ |
| $\zeta : 0.5$ |
| $\gamma : 2 \times 10^{-5}$ |
| initialization variance for $\Psi$: 0.05 |
| number of restarts for coefficient inference: 1 |
| nearest neighbor count: 5 |
| latent scale: 30 |

Table 6: Training parameters for the fine-tuning phase of the Fashion-MNIST experiment

| Fashion-MNIST Fine-tuning Parameters |
|---|
| batch size: 200 |
| transport operator training epochs: 150 |
| $lr_{\text{net}} : 10^{-4}$ |
| $lr_{\Psi} : 10^{-3}$ |
| $\zeta : 0.5$ |
| $\gamma : 2 \times 10^{-6}$ |
| $\lambda$: 0.75 |
| number of network update steps: 50 |
| number of $\Psi$ update steps: 50 |

Table 7: Training parameters for the Fashion-MNIST Coefficient Encoder

| Fashion-MNIST Coefficient Encoder Parameters |
|---|
| batch size: 200 |
| training epochs: 300 |
| lr: $10^{-3}$ |
| $\zeta_{\text{prior}} : 0.5$ |
| $\lambda_{\text{kl}}$: 0.5 |
| coefficient spread scale: 0.1 |
| classifier domain: latent |

## I  FASHION MNIST EXPERIMENT ADDITIONAL RESULTS

Here we show additional experimental details and results for the Fashion-MNIST experiment. Fig. 14 shows the magnitude of all 16 operators after the fine-tuning phase. Six of the operators had their magnitudes reduced to zero.

To visualize how the use of the transport operators varies over the latent space, we generate an Isomap embedding (Tenenbaum et al., 2000) of latent vectors and color each point by the encoded scale parameter for coefficients associated with each of the transport operators. Fig. 15a shows these embeddings for Fashion-MNIST data. Each operator has regions of the latent space where their use is concentrated. Fig. 15b shows the average coefficient scale weights for each class (rows)

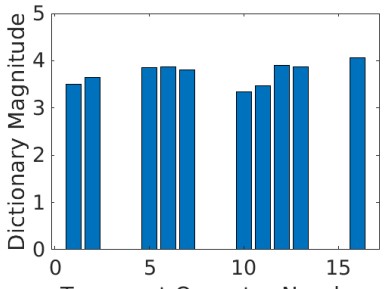

Figure 14: The resulting magnitude of the learned operators after fine-tuning the MAE.

and each transport operator (columns) for Fashion-MNIST. There are some classes like trouser and sandal (classes 1 and 5) which have large encoded coefficient scale weights for most of the transport operators. This means they are robust to many natural transformations. Other classes like coat and shirt (classes 4 and 6) have smaller encoded coefficient scale weights which means they are sensitive to many transformations. The images to the right in Fig. 15b show transport operators being applied to samples with high encoded scale weights (in a yellow box) and samples with low encoded scale weights (in a blue box). Fig. 16 shows the paths generated by non-zero transport operators trained on Fashion-MNIST data.

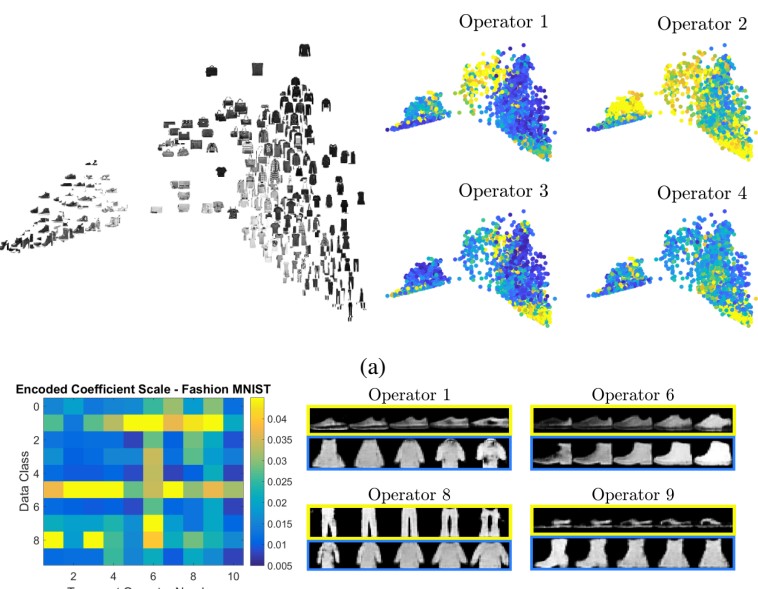

Figure 15: (a) Visualizations of the encoded coefficient scale weights. The leftmost image shows an Isomap embedding of the latent vectors with input images overlaid. The scatter plots on the right show the same Isomap embedding colored by the encoded coefficient scale weights for several operators (yellow indicates large scale weights and blue small scale weights). We see operators whose use is localized in regions of the manifold space. (b) The average coefficient scale weights for each class on each transport operator for Fashion-MNIST. High scale weights for a given operator (yellow) indicate it can be applied to a given class without easily changing identity. The images on the right show examples of the operators applied to classes with high encoded scale weights (in the top yellow boxes) and classes with low encoded scale weights (in bottom blue boxes). The examples with low coefficient scale weights change classes more easily than other examples.

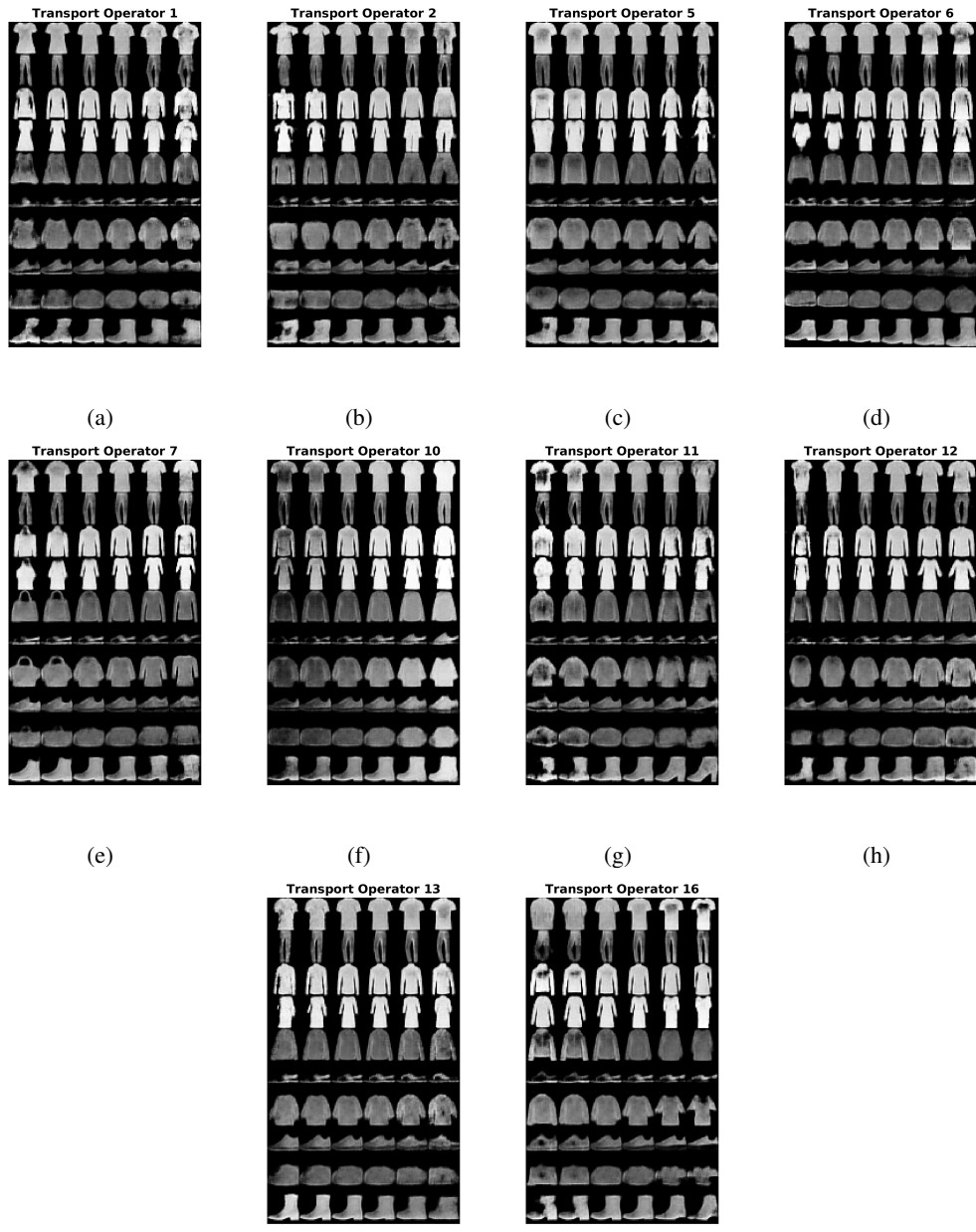

Figure 16: Paths generated by all non-zero transport operators trained on the Fashion-MNIST dataset. Images in the middle column of the image block are the reconstructed inputs and images to the right and left are images decoded from transformed latent vectors in positive and negative directions, respectively

## J CELEBA EXPERIMENT DETAILS

The CelebA dataset is publicly available for non-commercial research purposes. We split the CelebA dataset into training and testing sets. The training set contains the first 150,000 images accompanied with the entire test set. The input images are normalized so their pixel values are between 0 and 1. The network architecture used for the autoencoder is shown in Table 8. The training parameters for the transport operator training phase and the fine-tuning phase are shown in Table 9.

The attribute classifier is a ResNet-18 model adapted from https://github.com/d-li14/face-attribute-prediction with 16 classifier heads after the layer with 512 hidden units. Each classifier head corresponds to a single attribute and is modeled as a fully-connected linear layer with 256 hidden units, followed by batch normalization, dropout, and ReLU layers. Afterwards, two logits are output for a 0/1 prediction for each classifier output. The training procedure, including the dynamic loss weighting, follows (Mao et al., 2020).

Table 8: Network Architecture for CelebA Experiments

| Encoder Network | Decoder Network |
|---|---|
| Input $\in \mathbb{R}^{64 \times 64}$ | Input $\in \mathbb{R}^{32}$ |
| conv: chan: 32 , kern: 4, stride: 2, pad: 1 | Linear: 80,000 Units |
| BatchNorm: feat: 32 | ReLU |
| ReLU | convTranpose: chan: 256, kern: 3, stride: 1, pad: 0 |
| conv: chan: 64, kern: 4, stride: 2, pad: 1 | BatchNorm: feat: 256 |
| BatchNorm: feat: 64 | ReLU |
| ReLU | convTranpose: chann: 256, kern: 3, stride: 1, pad: 0 |
| conv: chan: 128, kern: 3, stride: 2, pad: 1 | BatchNorm: feat: 256 |
| BatchNorm: feat: 128 | ReLU |
| ReLU | convTranpose: chan: 256, kernel: 3, stride: 1, pad: 1 |
| conv: chan: 256, kern: 3, stride: 1, pad: 1 | BatchNorm: feat: 256 |
| BatchNorm: feat: 128 | ReLU |
| ReLU | convTranpose: chan: 128, kernel: 3, stride: 1, pad: 1 |
| conv: chan: 256, kern: 4, stride: 2, pad: 1 | BatchNorm: feat: 128 |
| BatchNorm: feat: 256 | ReLU |
| ReLU | convTranpose: chan: 128, kernel: 3, stride: 1, pad: 0 |
| conv: chan: 128, kern: 4, stride: 2, pad: 1 | BatchNorm: feat: 128 |
| BatchNorm: feat: 128 | ReLU |
| ReLU | convTranpose: chan: 3, kernel: 4, stride: 2, pad: 0 |
| Linear: 32 Units | Sigmoid |

Table 9: Training parameters for the CelebA experiment

| CelebA Transport Operator Training Parameters |
|---|
| batch size: 500 |
| autoencoder training epochs: 300 |
| transport operator training epochs: 50 |
| latent space dimension ($z_{dim}$): 32 |
| $M$ : 40 |
| $lr_{\text{net}} : 10^{-4}$ |
| $lr_{\Psi} : 10^{-3}$ |
| $\zeta : 1.5$ |
| $\gamma : 1 \times 10^{-5}$ |
| initialization variance for $\Psi$: 0.05 |
| number of restarts for coefficient inference: 1 |
| nearest neighbor count: 5 |
| latent scale: 2 |

| CelebA Fine-tuning Parameters |
|---|
| batch size: 500 |
| fine-tuning epochs: 10 |
| $lr_{\text{net}} : 10^{-4}$ |
| $lr_{\Psi} : 10^{-3}$ |
| $\zeta : 0.8$ |
| $\gamma : 5 \times 10^{-7}$ |
| $\lambda$: 0.75 |
| number of network update steps: 50 |
| number of $\Psi$ update steps: 50 |

## K  CELEBA EXPERIMENT ADDITIONAL RESULTS

Here we show additional experimental results for the CelebA experiment. Fig. 17 shows the paths generated by the 40 transport operators trained on celebA data. Each row represents a different operator acting on the same input image. Images in the middle column of the image block are the reconstructed inputs and images to the right and left are images decoded from transformed latent vectors in positive and negative directions, respectively. Fig. 18 shows the classification outputs of the attribute classifier for example transport operators. These two operators that vary smiling, beard, and sunglasses attributes.

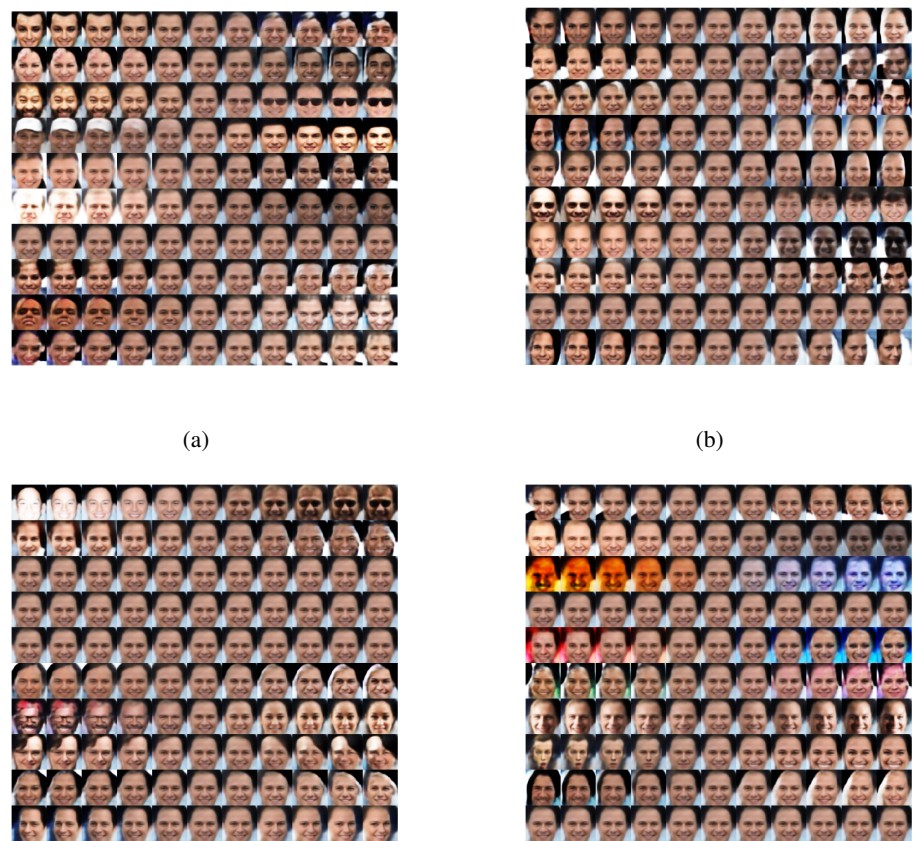

(a)

(b)

(c)

(d)

Figure 17: Paths generated by all 40 transport operators trained on the CelebA dataset. Images in the middle column of the image block are the reconstructed inputs and images to the right and left are images decoded from transformed latent vectors in positive and negative directions, respectively

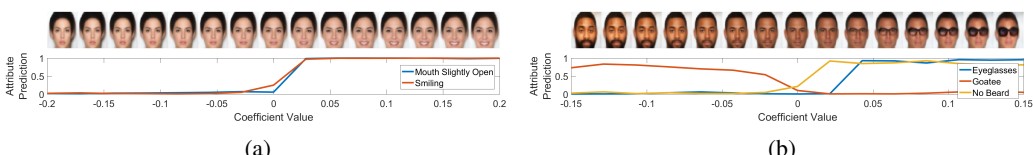

(a)

(b)

Figure 18: Paths generated by the application of two learned transport operators with the associated attribute classifier probability outputs for the transformed images. Our model learns operators that correspond to meaningful CelebA dataset attributes like (a) smiling (b) sunglasses/beard

Fig. 19 shows an interesting feature in our learned operators – many of them generate cyclic paths that begin and end at nearly the same point. Also, in these cyclic sequences, the transformations seem to lead to a change in gender. The image sequence in Fig. 19a shows the path generated by a single operator. The image in the middle is the reconstructed input image and the images to the left and right are the paths generated by negative and positive coefficients respectively. This operator changes the hairline and quantity of bangs. As we apply the transport operator with a negative coefficient (to the left of center), the woman gains bangs and then becomes a man with a moustache on the far left of the image. As we apply the transport operator with a positive coefficient (to the right of center), the woman's forehead gets higher and then she becomes a man with a high forehead and eventually, on the far right the woman transforms into a man with bangs, similar to the man on the far left. The similarity between the generated images on the far left and far right is notable

because this indicates the transformation path is nearly closed. Fig. 19b shows the change in five of the 32 latent dimensions over the generated path. These paths have a nearly cyclic structure.

This is particularly interesting because in (Connor & Rozell, 2020) they learn closed transformation paths by selecting point pairs on known cyclic transformation paths and highlight the benefit of the transport operator model for representing these types of paths. In this case, we learn this cyclic path with only perceptual point pair supervision. Additionally, this identifies a benefit of the transport operator approach over other models of the manifold in a neural network latent space. We can learn nonlinear paths that keep the generated points in the same neighborhood in the latent space without extending to infinity which is inevitable when linear paths that are used to represent natural transformations.

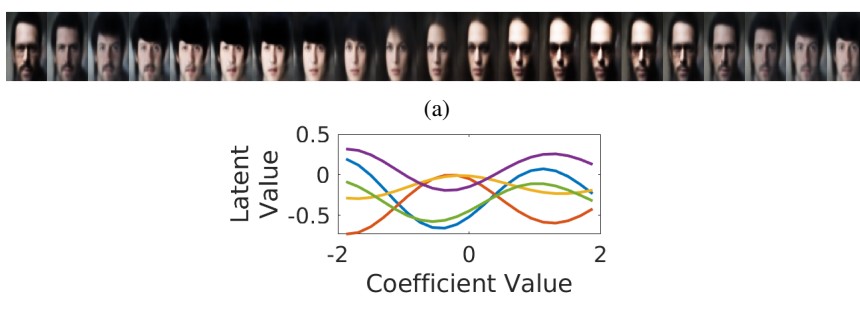

(a)

(b)

Figure 19: An example of an operator that generates a nearly cyclic path in the latent space (a) Image outputs along a path generated by a single learned operator. The images on the far left and far right look similar which indicates that this operator generates a nearly closed path that begins and ends at the same point. (b) Paths of five of the 32 latent dimensions as the learned operator is applied to them. This again highlights the cyclic nature of the transport operator generated paths.

