# OpenReview forum: "Learning Identity-Preserving Transformations on Data Manifolds"
_ICLR.cc/2022/Conference — ICLR 2022 Submitted_

### Official Review · Reviewer_98g7 · 2021-11-04

**Correctness:** 3
**Technical Novelty And Significance:** 2
**Empirical Novelty And Significance:** 2
**Recommendation:** 5
**Confidence:** 4

**Main Review:**

The idea of using features from classification networks is quite interesting. The proposed way of learning local transport also makes sense. However,  the way of MAE is an ill-posed decomposition of encoding-decoding and group actions. The additional step of combining MAE together with another encoder network for local operation makes the decomposition even more ill-posed. It would be nice to have a better explanation of why the desired decomposition (i.e. latent variables, group action, local action regions) can be obtained.

**Summary Of The Paper:**

The manuscript proposes a method for learning group action on data using features from classification networks. The idea is natural and easy to follow.

**Summary Of The Review:**

In fact, I found the proposed method is quite interesting.  A better explanation of why the proposed ill-posed decomposition can provide the desired solution would make the idea more convincing.

---

> ### Author Response · Authors · 2021-11-16
> **Reviewer 98g7 Initial Response**
>
> We thank the reviewer for saying that our proposed methods are interesting. We read the reviewer’s question as asking whether our proposed methods provide any guarantees in learning the data manifold given the decomposition inside of a deep neural network. Although our results are mostly empirical, we present several training techniques that can help practitioners apply the transport operator model to a wide variety of success. We find that the decomposition inside of a deep neural network (that is trained to extract latent factors through a reconstruction objective) allows us to bypass the curse of dimensionality from learning the Lie group operators in the pixel space. This allows us to learn several interesting transformations that would be challenging to learn in the pixel space. With regards to learning the data manifold, the transformations learned are determined by the point pairs selected during training. When transformation labels are not available to select point pairs, we empirically show in the appendix that selecting nearest neighbors in the latent space of a pre-trained classifier leads to more stable operators that coincide with the smooth, semantically meaningful transformations we show in the main text. Please let us know if we have misinterpreted this main concern.

---

### Official Review · Reviewer_qg7U · 2021-11-04

**Correctness:** 3
**Technical Novelty And Significance:** 3
**Empirical Novelty And Significance:** 4
**Recommendation:** 5
**Confidence:** 4

**Main Review:**

#### Strengths
The authors consider the important and challenging task of learning identity preserving transformations. Much of the previous work considers physical transformations (e.g., rotations, scaling) rather than semantic transformations which is arguable more challenging due to the lack of transformation labels (consider citing exception [1]). The challenge is compounded further by the presence of multiple classes in the dataset since not all transformations make sense for all classes. The proposed solutions for these challenges is interesting and the empirical analysis shows that they are also effective.

[1] Ham, Jihun, and Daniel D. Lee. "Separating pose and expression in face images: a manifold learning approach." Neural Information Processing-Letters and Reviews 11.4 (2007): 91-100.

#### Weaknesses
It is difficult for me to reconcile the use of Lie operators which assume some smoothness in the data manifold and the statement: “However, semantic transformations of interest may not be exhibited through pixel similarity nor through unsupervised autoencoder feature similarity”. Moreover, the NN samples shown in Figure 9 seem to say otherwise. Specifically, the hair style between Latent NN examples appears more similar than those of the Perceptual NN.

I'm not convinced the use of CAE and beta-VAE as baselines is appropriate. Comparison against more recent work in disentangled representation learning (e.g., [2]) would be more convincing. I would have also liked to see a comparison against another approach that uses Lie operators (e.g., [1, 3])

A common dataset for analysis of disentangled representation learning algorithms is [4]. I would have liked to see results on this dataset.

[2] Yu Deng, Jiaolong Yang, Dong Chen, Fang Wen, Xin Tong; Proceedings of the IEEE/CVF Conference on Computer Vision and Pattern Recognition (CVPR), 2020, pp. 5154-5163

[3] Connor, Marissa, and Christopher Rozell. "Representing closed transformation paths in encoded network latent space." Proceedings of the AAAI Conference on Artificial Intelligence. Vol. 34. No. 04. 2020.

[4] Locatello, Francesco, et al. "Challenging common assumptions in the unsupervised learning of disentangled representations." international conference on machine learning. PMLR, 2019.


**Summary Of The Paper:**

The authors introduce an approach for learning identity preserving transformations from data. First the low-dimensional manifold structure of the data is learned using an autoencoding neural network, then the transformations (the Lie group operators and their coefficients of combination) that map between perceptually similar image pairs are learned. The main contribution of the paper is that the learned operators can transform the data semantically. This is a challenging since 1) semantic transformation labels are not usually available; and 2) not all semantic transformations make sense for all elements of the dataset. The authors address these challenges using two auxiliary networks. The first identifies perceptually similar image pairs in the dataset and the second identifies where particular transformations are likely to be used.

**Summary Of The Review:**

The proposed work is interesting and addresses an important challenge in representation learning. I'm not convinced the baselines or datasets used in the empirical analysis are most appropriate for this work.

---

> ### Author Response · Authors · 2021-11-16
> **Reviewer qg7U Initial Response**
>
> Thank you for your substantial feedback. We appreciate your comments about our solutions being interesting and our empirical analysis being effective. We thank the reviewer for the additional citations and will be sure to include them in our revision.
>
> We would like to motivate our choice of the Lie group model in representing smooth image transformations. Although humans are adept at envisioning smooth variations between images, these variations often cannot be represented by linear paths between the pixel representations of images. In some cases linear interpolation in an unsupervised feature space may result in smooth variations, but it is not guaranteed without encouraging a specific latent structure. In our work, we train an autoencoder network to find a latent space that extracts latent factors that can be transformed and decoded for smooth image variations. This allows us to bypass the computational requirements of working in the pixel space. We show empirically in Appendix E that our perceptual similarity method for selecting point pairs results in more stable transformations in the latent space that can extend farther than using nearest neighbors in the latent space. We demonstrate that the latent transformations can be decoded for smooth transformations in the image space in Figures 1-3. By using this approach rather than defining linear transformation paths in the latent space  (as is done with many disentanglement methods), we are able to identify interesting transformations that can be used to more effectively extrapolate transformation paths (as we show in Figures 3 and 4)
>
> Expanding on this last point, we believe that the goal of our work differs from that in the disentanglement literature. Rather than identifying independent factors of variation in the data, we wish to learn non-linear operators that correspond to transformations on the data manifold. These two goals do not necessarily result in the same transformations or factors of variation.  We choose the CAE and beta-VAE as two alternate methods of identifying natural transformation directions within datasets without transformation labels.  Although it would be ideal to compare to other Lie group operator methods, many existing methods focus a subset of known transformations with predefined Lie group operators and/or rely on transformation supervision when learning the Lie group operators. We note that many datasets of interest do not have transformational supervision which motivates the formulation of our new method for effectively learning unlabeled data transformations. We agree that a deeper  discussion about the relationship between our work and previous Lie group operators methods would add to our paper and we will include it in our revision.

---

### Official Review · Reviewer_uMAL · 2021-11-04

**Correctness:** 3
**Technical Novelty And Significance:** 2
**Empirical Novelty And Significance:** 2
**Recommendation:** 6
**Confidence:** 3

**Main Review:**


My main concern is with the backbone chosen (MAE) in this paper and its training, to demonstrate the idea of learning manifold transformation effectively. Three-stage training phase for MAE is already something and on top of that, this paper introduces to exploit a pretrained Resnet classifier and a coefficient encoder network with Laplace distribution prior on the coefficients. As a result, the overall approach errs on the side of being a bit complicated.

Presentation:
I found page 5 quite hard to follow. As a result, I don't understand the fine tuning phase described as such on page 6. 'The fine-tuning phase adapts the autoencoder latent space to fit the learned transformations in the event that that pretrained latent space does not match the desired data manifold' I understand this without coefficient encoder network but not sure what is being matched after the introduction of coefficient encoder network.

Also, why name Section 3 as Methods? if named MAE, it will help to delineate your contributions from MAE clearly.

The experimental setup is ok although I don't understand why choose only two baselines, the latest being from 2017.

**Summary Of The Paper:**

This paper proposes to learn natural transformations in datasets, with manifold auto encoder (MAE), where the underlying identity-preserving transformations are not easily identifiable. By using Lie group operator, this problem reduces to learning paths/motion in the latent space of MAE. The main challenge in training MAE is how to choose a transformation pair, a point and its identity preserving transformed counterpart, in an unsupervised way. This paper introduces the idea of using penultimate layer  of a pretrained classifier on Imagenet to choose such point pairs and learn such MAE more effectively. The proposed method is validated on three datasets.

**Summary Of The Review:**

I give marginally above for now but I will revisit it based on other reviews and discussions. The problem of learning manifold transformation is well motivated. Experiments seem ok. The idea to use pre-trained classifier to fit MAE latent space traversal is sensible.

---

> ### Author Response · Authors · 2021-11-16
> **Reviewer uMAL Initial Response**
>
> We thank the reviewer for their thoughtful comments. We appreciate the assessment that the problem of learning manifold transformations is well motivated.
>
> We are currently working on a rebuttal revision where we will clarify our methodology and training procedure per your suggestions. Stated here briefly, our initial training phase consists of training an auto-encoder with a reconstruction objective to extract latent factors that represent our image dataset. Afterwards, we freeze the auto-encoder weights and train our transport operator model in the latent space as an initial fit. In the final phase, we combine the transport operator and reconstruction training objectives and alternate between gradient steps on the auto-encoder weights and the transport operator weights. We find this third stage is critical for learning nonlinear manifold paths that can be extended to represent natural transformations. Although this training procedure uses multiple steps, we find in our experiments it leads to an increase in stability over a single training phase. Addressing the complexity of the training setup, we would like to point out to the reviewer that many similar methods use a combination of networks and feature representations [1, 2, 3]. We ask the reviewer to let us know if there is additional content that needs clarification (such as the coefficient encoder).
>
> With regards to our baselines, our goal is to select similar methods that aim to learn the data manifold and identify semantic transformations of interest in the data set. Although it would be ideal to compare to other Lie group operator methods, many existing methods focus a subset of known transformations with predefined Lie group operators and/or rely on transformation supervision when learning the Lie group operators. We note that many datasets of interest do not have transformational supervision which motivates the formulation of our new method for effectively learning unlabeled data transformations. We use the CAE (approximates manifold tangent) and beta-VAE (disentangles factors of data variation) to show two alternate methods of identifying natural transformation directions within datasets without transformation labels. We note that the goal of our work (learning nonlinear representations of manifold transformations) is different from the disentanglement literature which aims to identify independently varying factors of variation. However, we show in Figure 1 that our learned transformations are semantically similar to the variations identified using beta-VAE, which aims to disentangle, without harming reconstruction performance. We acknowledge that further discussion of this point is needed and will include further comparison to disentanglement in our revision.
>
> [1] D. Sussillo, R. Jozefowicz, L. F. Abbott, en C. Pandarinath, “LFADS - Latent Factor Analysis via Dynamical Systems”, arXiv [cs.LG]. 2016.
>
> [2] Y. Rubanova, R. T. Q. Chen, en D. Duvenaud, “Latent ODEs for Irregularly-Sampled Time Series”, arXiv [cs.LG]. 2019.
>
> [3] C.-W. Kuo, C.-Y. Ma, J.-B. Huang, en Z. Kira, “FeatMatch: Feature-Based Augmentation for Semi-Supervised Learning”, arXiv [cs.CV]. 2020.

---

### Public Comment · ~Romain_Cosentino2 · 2021-11-10
**extra refs**

Dear authors,

Nice and interesting paper.

1) There is a recent work, also about learning Lie group generators for autoencoders, that is similar to your work, it would be interesting to discuss the differences between the two, the link is: https://msml21.github.io/papers/id49.pdf.

2) You might be interested in checking and citing this paper: https://papers.nips.cc/paper/2017/file/86a1793f65aeef4aeef4b479fc9b2bca-Paper.pdf, for curiosity and to maybe speed up even more the computations!

---

### Decision · Program_Chairs · 2022-01-20

**Decision:**

Reject

**Comment:**

The paper proposes a method for learning identity-preserving transformations through a set of learned Lie group operators. It builds upon previous work ( (Connor & Rozell, 2020; Connor et al., 2021) addressing two points: (i) how to select semantically related pairs of points, (ii) how to identify which operators are appropriate for a given local region of the manifold. Authors use nearest neighbors computed via the penultimate layers of a pretrained network to address (i), and learn a separate network q(c|z) that predicts the coefficients given a latent input z. Reviewers have two main concerns with the paper -- limited novelty over earlier work that learns the Lie group operators; and the complicated nature of the method which needs training in three stages and uses a pretrained ResNet for finding nearest neighbors. Lack of comparison with relevant baselines is also pointed out by the reviewers. Given these issues, the paper is unfortunately not suitable for publication in ICLR at this point.